# MAP TO OPTIMAL: ADAPTING GRAPH OUT-OF-DISTRIBUTION IN TEST TIME

## ABSTRACT

Based on topological proximity message passing, graph neural networks (GNNs) can quickly model data patterns on graphs. However, at test time, when the node feature and topological structure of the graph data are out-of-distribution (OOD), the performance of pre-trained GNNs will be hindered. Existing test-time methods either fine-tune the pre-trained model or overlook the discrepancy between the prior knowledge in pre-trained models and the test graph. We propose a novel self-supervised test-time adaptation paradigm (GOAT[1]), through graph augmentation-to-augmentation strategy, that enables a simple adapter to mitigate the distribution gap of training data and test-time data. GOAT reduces generalization error for node classification in various pre-trained settings through experiments on six benchmark datasets spanning three distinct real-world OOD scenarios. Remarkably, GOAT outperforms state-of-the-art test-time methods, and our empirical study further demonstrates the interpretability of the OOD representation generated from our method.

## 1 INTRODUCTION

Graph pre-training has emerged as a powerful technique for preserving information from large-scale upstream data (Kipf and Welling, 2016; Hamilton et al., 2017; Veličković et al., 2018; Hu et al., 2019; Lu et al., 2021), enabling graph neural networks (GNNs) to learn rich representations that can be transferred to various downstream graph tasks. Whereas, the effectiveness of pre-trained GNNs is often hampered by distribution shifts (Zhu et al., 2020; Wu et al., 2021a; Koh et al., 2021; Yehudai et al., 2021; Li et al., 2022a), especially in real-world scenarios where the test data is out-of-distribution (OOD) and labels are unavailable. This poses a significant challenge for the practical application of GNNs, as their performance tends to deteriorate severely under such distribution shifts. Intuitively, in Table 1, the pre-trained GNN's performance degrades when the OOD test data evolves through time. Therefore, adapting pre-trained GNNs to test graphs with no labels and OOD data is crucial.

To address the issue of distribution shift, various approaches have been proposed, such as invariant risk minimization (Arjovsky et al., 2020; Wu et al., 2022), domain-invariant learning (Muandet et al., 2013), and invariant representation learning (Wu et al., 2021b; Li et al., 2022b). These methods aim to learn representations that are robust to distribution shifts by explicitly optimizing for invariant variables across different domains or environments. However, a common limitation of these approaches is their reliance on labeled data from the target domain during training, which may not always be available in real-world scenarios. Moreover, these methods are designed to be applied during the training phase and do not deal with the challenges of adapting pre-trained GNNs to OOD data at test time, when access to labeled data is often limited or nonexistent.

On the other hand, existing graph test-time adaptation approaches, such as reconstructing the classifier head of GNNs (Wang et al., 2022) or fine-tuning the entire model's parameters (Zhang et al., 2024; Wang et al.), trying to focus on leveraging the generalization ability of the pre-trained model. These model-centric approaches face difficulties in handling OOD scenarios due to their reliance on learned parameters that may not generalize well to unseen distributions (Hendrycks and Dietterich, 2019; Arjovsky et al., 2020). Furthermore, fine-tuning the model's parameters often requires significant computational resources, making it challenging or even infeasible in resource-constrained

---

[1]GOAT is at `https://anonymous.4open.science/r/GOAT-5C0E`

Table 1: The showcase indicates a significant decrease in the ERM pre-trained GCN's node classification performance on the OGB-ArXiv (Hu et al., 2020) and Elliptic (Pareja et al., 2020) datasets in an OOD setting where graph data is generated from different time environments. For OGB-ArXiv, the year ranges from before 2011 to 2020; for Elliptic, from the $7^{th}$ snapshot (when the dark market crackdown occurred) to the $49^{th}$ snapshot. Performance degrades noticeably during validation and testing as time progresses.

| Dataset | OGB-ArXiv | | | Elliptic | | |
|---|---|---|---|---|---|---|
| Description | Open-world dataset of academic papers, the graph evolves as new papers are cited. | | | A dataset of transactions labeled as licit or illicit, influenced by market conditions. | | |
| Split | Year Slice | Accuracy | Degrade | Time Slice | F1 score | Degrade |
| **Train** | before - 2011 | 47.88% | | $7^{th}$ - $11^{th}$ | 90.12% | |
| **Val** | 2011 - 2014 | 44.46% | -9.96% | $12^{th}$ - $17^{th}$ | 78.75% | -39.17% |
| **Test** | 2014 - 2020 | 38.92% | | $17^{th}$ - $49^{th}$ | 50.95% | |

environments. Meanwhile, although data-centric methods (Jin et al., 2022; Chen et al., 2022b; Fang et al., 2024) have shown potential in graph test-time adaptation through fine-tuning the OOD test graph, there still exists the limitation of overlooking the discrepancy between the prior knowledge in pre-trained models and the test graph and lack of interpretability.

Recently, parameter-efficient prompt-based methods (Sun et al., 2022; Liu et al., 2023b; Sun et al., 2023; Yu et al., 2023; Fang et al., 2024; Lee et al., 2024) have shown promise in bridging the gap between pre-trained models on upstream tasks and downstream tasks in the graph domain. Existing prompt methods aim to adapt pre-trained GNNs to new tasks by designing task-specific prompts that guide the model to generate relevant representations. Nevertheless, prompt-based methods still rely on labeled data to adapt to new tasks and the designed prompts do not explicitly address the data-centric differences between the upstream training set distribution and the test data distribution.

**Present work.** We propose **G**raph **O**ut-of-distribution **A**ugmentation-to-augmentaion adaptation in **T**est time (GOAT), a novel self-supervised test-time tuning strategy that allows the model to dynamically adapt to unknown test distributions without requiring access to source training data or training details. (1) A key contribution of GOAT is considering the symmetry and consistency of the OOD test graph by collaborating the point estimation with the commutativity of the representation of GNN and the adapter. (2) In addition to achieving parameter efficiency and interpretability, GOAT introduces a topology-aware feature bias adapter, which acts as a data-centric distribution mapping, indicating the OOD degree of the test graph. (3) We conduct extensive experiments on multiple real-world datasets, demonstrating the superiority of GOAT in handling distribution shifts on different backbones and comparing them to state-of-the-art baselines.

## 2 RELATED WORK

**Distribution Shift on Graphs.** Graph-structured data often exhibits out-of-distribution (OOD) phenomena (Li et al., 2022a; Song and Wang, 2022). To tackle this challenge, researchers have proposed methods for learning invariant representations (Muandet et al., 2013; Arjovsky et al., 2020; Wu et al., 2021b; 2022; Li et al., 2022b), generalizing pre-trained GNNs (Hu et al., 2019; Zhao et al., 2021; Zhu et al., 2021; Li et al., 2022a; Song and Wang, 2022; Guo et al., 2023; Shen et al., 2023), detecting OOD instances (Li et al., 2022c; Bazhenov et al., 2024; Huang et al., 2024). Most of these approaches often require access to multiple source domains, rely on specific model architectures, or train-time paradigms, or may lead to performance degradation. For a thorough review, we refer the readers to a recent survey (Wu et al., 2024).

**Graph Test-time Adaptation.** Graph test-time adaptation (TTA), as first introduced by Chen et al. (2022b), aims to adapt pre-trained models to the test distribution without requiring labeled data or modifying the model's parameters (Sun et al., 2020; Chen et al., 2022b). Methods like Tent (Wang et al.), MEMO(Zhang et al., 2022), GTrans (Jin et al., 2022), GT3 (Wang et al., 2022), and GraphCTA (Zhang et al., 2024) have been proposed, but they have limitations such as reliance on specific architectures, over-smoothing, or lack of interpretability. These works inspire us to introduce self-supervised tasks at test time to improve the robustness and generalization performance

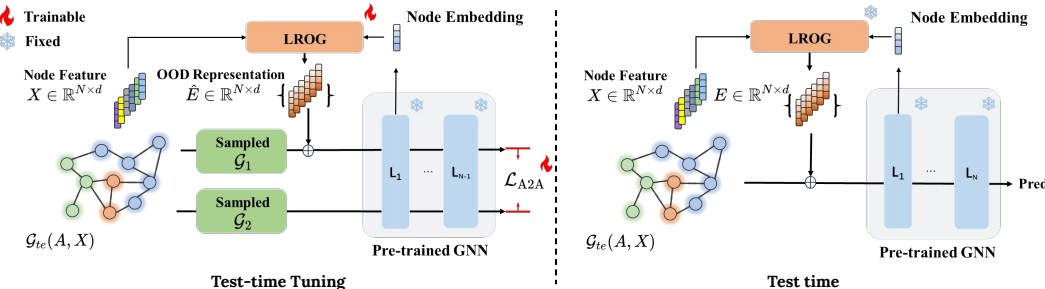

Figure 1: Overview of the proposed method GOAT under two sample views. In **test-time tuning**, a single test graph's augmented views sampled under the same environment are passed through the GNN with their respective OOD representation added on the node feature. A self-supervised loss $\mathcal{L}_{\text{A2A}}$ is applied to refine the adapter LROG supervising the alignment between the embeddings of the one augmented input w/ additional bias and another one w/o. When it comes to the **test time**, all parameters will be frozen and the OOD representation $\boldsymbol{E}$ will be added and pass through the GNN.

of GNNs. Our GOAT focuses on the representation space and introduces a novel self-supervised test-time tuning strategy.

**Prompt Tuning & Adapter.** Prompting reformulates downstream tasks by introducing prompts to narrow the gap with pre-training objectives, which has gained attention in NLP (Liu et al., 2023a). Continuous prompts (Liu et al., 2021), pre-fix prompts (Rao et al., 2022; Shu et al., 2022; Zhou et al., 2022b;a), and low-rank adaptation (Hu et al., 2021) have been explored to adapt to downstream tasks and compress tuning parameters. Efforts have been made to extend prompt tuning to graph neural networks by introducing additional bias or scaling on the parameters, e.g. Sun et al. (2022); Liu et al. (2023b); Sun et al. (2023); Yu et al. (2023); Fang et al. (2024); Lee et al. (2024), enabling effective knowledge transfer. Instead of altering the model design or specific training method, our adapter is only applied at test time and converts the topological information into the node feature space.

## 3 METHODOLOGY

In this section, we delve into the **G**raph **O**ut-of-distribution **A**ugmentation-to-augmentaion in **T**est-time (GOAT) model in Figure 1 to elucidate its underlying intuitions and technical intricacies. A figure illustration of why our GOAT paradigm works can be found in Appendix B.

### 3.1 PROBLEM FORMULATION

**Graph Pre-training.** For any graph-structure data, let $\mathcal{G} = \{\boldsymbol{A}, \boldsymbol{X}, \boldsymbol{Y}\}$ denote the test graph, where $\boldsymbol{A} \in \{0,1\}^{N \times N}$ is the adjacency matrix, $N$ is the number of nodes, $\boldsymbol{X} \in \mathbb{R}^{N \times d_0}$ is the $d$-dimensional node feature matrix, $\boldsymbol{Y} \in \mathbb{R}^{N \times C}$ is the one-hot encoded label set of the $N$ nodes, and the number of $N$-node classes is $C$.

**Assumption 1.** *Environment is the condition that generates graph.* Assume that graph $\mathbf{G}$ and environment $\mathbf{e}$ are random variables. There exists an environment set $\mathcal{E} = \{\boldsymbol{e}_1, \boldsymbol{e}_2, ..., \boldsymbol{e}_i\}$ represents any graph generated via distribution $\mathcal{G} \sim p(\mathbf{G}|\mathbf{e} = \boldsymbol{e}_i)$.

Let $\widetilde{D}_{tr} = \{\mathcal{G} \sim p(\mathbf{G}|\mathbf{e} = \boldsymbol{e}_1)\}$ be the pre-training graph dataset generated from the training environment $\boldsymbol{e}_1$, noted that $\widetilde{D}_{tr}$ could be a set includes a single graph or multiple graphs. We would have a pre-trained GNN $f_\theta$ optimized with any loss function $\mathcal{L}_{tr}(\cdot, \cdot)$ on $\mathcal{G}_{tr} \in \widetilde{D}_{tr}$ by following equation:

$$\theta^* = \arg\min_\theta \int \mathcal{L}_{tr}(f_\theta(\mathcal{G}_{tr}), \boldsymbol{Y}_{tr})\, p(\mathbf{G}|\mathbf{e} = \boldsymbol{e}_1)\, d\mathbf{G}. \tag{1}$$

**Data-centric Graph Test-time Adaptation(DGTTA).** At test time, we have graph data $\mathcal{G}_{te}, \hat{\mathcal{G}}_{te} \sim p_{1 \le i}(\mathbf{G}|\mathbf{e} = \boldsymbol{e}_i)$, but no access to the corresponding labels $\boldsymbol{Y}_{te}$. We formulate the DGTTA problem

as a point estimation problem, where $\theta$ is fixed by $\theta^*$:

$$\arg\min_\psi \int \mathcal{L}_{te}(f_{\theta^*}(g_\psi(\mathcal{G}_{te})), f_{\theta^*}(\hat{\mathcal{G}}_{te}))\, p(\mathbf{G}|\mathbf{e} = \boldsymbol{e}_i)\, d\mathbf{G}, \tag{2}$$

where $g_\psi : \mathbb{G} \to \mathbb{G}$ is a graph transformation parameterized by $\psi$ and $\mathcal{L}_{te}$ is a self-supervised loss function. The optimal parameter $\psi^*$ minimizes the expected supervised loss $\mathcal{L}_{sup}(\cdot, \cdot)$:

$$\psi^* = \arg\min_\psi \mathcal{L}_{sup}(f_{\theta^*}(g_\psi(\mathcal{G}_{te})), \boldsymbol{Y}_{te}). \tag{3}$$

To further extend our framework, the self-supervised $\mathcal{L}_{te}$ in Eq.(2) can be reinterpreted as a path $\gamma$, that transforms one graph into another within their node embedding space, integral over graph space, the range of $g_\psi$, which facilitates a more nuanced understanding of how the graph representations evolve during the adaptation phase:

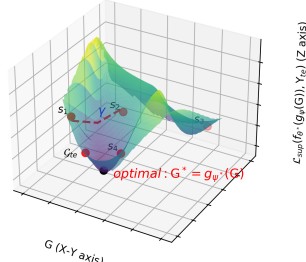

$$\arg\min_\psi \int \left\| \int \gamma\, ds \right\|_m^m p(\mathbf{G}|\mathbf{e} = \boldsymbol{e}_i)\, d\mathbf{G}, \tag{4}$$

where $s$ denotes continuous points on the path $\gamma$ due to different transformations, such as $g_\psi$ and identity mapping, of the input graph in the input graph space. The toy visualization in Figure 2 depicts the X-Y axes representing the graph space, while the Z-axis indicates the supervised loss values. $\mathbf{G}^*$ represents the optimal transformed graph of the test graph $\mathcal{G}_{te}$. Red nodes correspond to the loss values of different graphs, and the path between the red nodes illustrates the transformation pathway.

Figure 2: A toy visualization. An example of what $\gamma$, $s$, and the Eq.(4) are.

## 3.2 Graph Test-Time Out-of-Distribution Adapter

In this section, we introduce a novel approach to adapt pre-trained GNNs to out-of-distribution (OOD) data by using a Graph Test-time Distribution Adapter, instead of devising a specially designed prompt method on GNNs or simply adding a randomly initialized bias on the input test graph. Our adapter aims to fit a function playing the role of isomorphic mapping, capturing OOD representation, and reintegrating the modified test graph's node feature back into the GNN, thereby effectively representing distribution-shifted environments without the combinatorial explosion in edge search space (Jin et al., 2022; Zhang et al., 2024).

We introduce the following proposition to ensure that the additional learnable parameters on pre-trained GNN are reasonable and do not introduce an excessive learning burden.

**Proposition 1.** *Parameter-Efficient Graph Test-time Adaptation.* We define a function $g_\psi : \mathbb{G} \to \mathbb{G}$, parameterized by $\psi$, that modifies the graph structure within the learned parameter space of a pre-trained GNN model $f_{\theta^*}$. This function optimizes a given objective in Eq.(2). It should be ensured that the parameter count and computational complexity of $g_\psi$ should be significantly lower than those of $f_{\theta^*}$.

In addition, since the pre-trained $\theta^*$ are fixed at test time and the test graph is the focus of this paper, we drop the subscript in $\mathcal{G}_{te}$ and $f_{\theta^*}$ to simplify notations in the rest of the paper.

**Proposition 2.** Given a pre-trained GNN $f$, for each $\mathcal{G} \sim p(\mathbf{G}|\mathbf{e} = \boldsymbol{e}_i)$, where $\boldsymbol{e}_i \in \mathcal{E}$, there exists a specific matrix $\hat{\boldsymbol{E}}$ representing the OOD under $p(\mathbf{e} = \boldsymbol{e}_i|\boldsymbol{e}_1, f)$.

**Adapter Design.** We propose the Low-Rank Out-of-distribution Generalization (LROG) adapter. It's designed to handle *large-scale* graph complexities by introducing a low-rank attention mechanism that focuses on significant nodes, reducing irrelevant feature information and computational complexity, thereby facilitating the learning of invariant features across node dimensions.

The core idea is to use a low-rank projection to transform node features $\boldsymbol{X}$ and their embedding $\boldsymbol{H}$ into a lower-dimensional space that captures essential information across different OOD environments, which is crucial for maintaining model performance under distribution shifts. The adapter operation is formally expressed as:

$$g_\psi(\mathcal{G}) = \mathcal{G} \oplus \hat{\boldsymbol{E}} = (\boldsymbol{A}, \boldsymbol{X} + \hat{\boldsymbol{E}}), \tag{5}$$

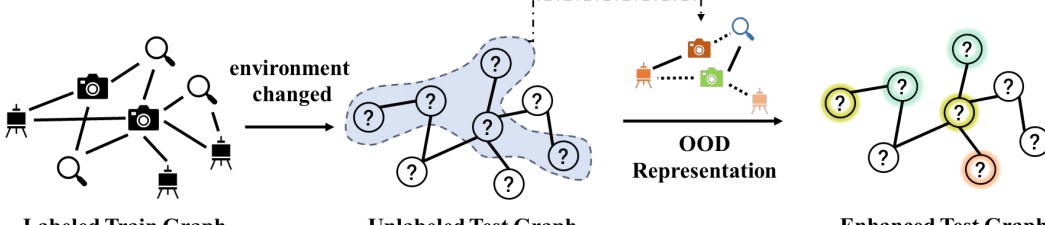

**Labeled Train Graph**   **Unlabeled Test Graph**   **Enhanced Test Graph**

Figure 3: An illustration of what LROG learns. A node classification task where we need to determine whether a node in a new graph represents a camera, lens, or tripod. The connections in the graph indicate co-purchases. However, during testing, due to economic environmental changes, the connections may decrease. The OOD representation generated from our LROG module can catch both the change in graph structures and feature distribution shifts in test time.

where the OOD representation $\hat{E} \in \mathbb{R}^{N \times d_0}$ is generated through a LROG as follows:

$$\hat{E} = \text{FFN}(\text{Softmax}\left(\frac{Q(K'K)^T}{\sqrt{d_{\text{attn}}}}\right)(V'V)). \tag{6}$$

Denoted that FFN($\cdot$) follows the design in Chen et al. (2022a). $Q$, $K$, and $V$ represent the linear transformations of query, key, and value, respectively:

• **Query $Q$:** The query matrix $Q$ is obtained by transforming the input node features $X \in \mathbb{R}^{N \times d_0}$ using a weight matrix $W_Q \in \mathbb{R}^{d_0 \times d_{\text{attn}}}$.

• **Key $K$ and Value $V$:** The key and value matrices are derived from the node representations $H^{(k)} \in \mathbb{R}^{N \times d_k}$, which encapsulate the aggregated neighborhood information up to the $k$-th layer. [2] They are computed using their respective weight matrices: $W_K, W_V \in \mathbb{R}^{d_k \times d_{\text{attn}}}$.

• **Low-Rank Projections $K'$ and $V'$:** $K', V' \in \mathbb{R}^{|n| \times N}$ are learned matrices that reduce the node-wise dimension of $K$ and $V$ to a lower-rank space, denoted by $|n| \times d_{\text{attn}}$. [3]

**Time Complexity**, assuming $k = 1$, is $O(Nd_{\text{attn}})$, where $N$ is the number of nodes and $d_{\text{attn}}$ is the LROG attention dimension.

**An instance of what LROG learns**. *"In recommender systems, a decline in users' purchasing power, influenced by economic conditions, may lead to a reduction in co-occurrence connections between different products."* This is a text description of the environment that generates the OOD distribution, which can be effectively encoded into the node feature space through a well-designed feature transformation, as illustrated in Figure 3. Specifically, the labeled train graph describes the product co-occurrence relations and the OOD unlabeled test graph changes due to the economic condition change. Changes in the environment can lead to different topological structures and shifts in feature distributions. The incorporation of such prior knowledge introduces an inductive bias that guides the GNN to learn in a direction more consistent with the prior knowledge, thereby accelerating convergence and improving the learning outcome.

### 3.3 AUGMENTATION-TO-AUGMENTATION UNVEILS DISCREPANCY

To further enhance the generation of LROG under distribution shifts, we propose an unsupervised loss $\mathcal{L}_{A2A}$ including both symmetric and consistent losses with regularization. Inspired by Glasserman and Ho (1990) that slight perturbations of the input can be used to approximate gradients, we also estimate the gradient of adapter $g_\psi$ by making small perturbations during the test time.

**Assumption 2.** Given a $\mathcal{G} \sim p(\mathbf{G}|\mathbf{e} = e_i)$, its augmented view dataset $\widetilde{D}_{aug} = \{\mathcal{G}', \mathcal{G}'', ..\}$ sampled during the test-time tuning with tiny disturbance according to same math statistic attributes, such as average degree, average node feature, or total edge numbers, etc., also share the same distribution with $\mathcal{G}$, i.e. $\widetilde{D}_{aug} \subseteq p(\mathbf{G}|\mathbf{e} = e_i)$.

---

[2] $k \geq 1$. Based on our Proposition 1, to reduce complexity, in our experiment, $k$=1.

[3] It should satisfy that $|n| \ll N$, as empirically proved in Sec.4.3.

Thus, it paves the way for LROG to learn the encapsulated representation of the test-time OOD.

By optimizing the objective in Eq.(2) with the squared $\mathcal{L}^2$ norm $\| \cdot \|^2$, the enhanced test graph's representation mapped by GNN ultimately aligns the mathematical expectation in this environment with the correct decision boundary learned during training.

**Symmetry.** The key idea is to utilize the adapter to minimize the discrepancy between the input test graph and its enhanced graph under the same environment $e_i$ in terms of their GNN mappings. By sampling $|v|$ augmented graphs $\mathcal{G}_v \sim p(\mathbf{G}|\mathbf{e} = e_i)$, we consider the paths $\gamma_{p,q}$ connecting the augmented graphs $\mathcal{G}_p$ and $\mathcal{G}_q(p, q \in \{1, ..., |v|\})$ in the input space. According to the fundamental theorem of calculus, the difference between the outputs of $f \circ g_\psi$, as estimation, and $f$, at $\mathcal{G}_p$ and $\mathcal{G}_q$, can be expressed as an integral of the gradient along the path $\gamma_{p,q}$:

$$f(g_\psi(\mathcal{G}_p)) - f(\mathcal{G}_q) = \int \gamma_{p,q} \nabla(f \circ g_\psi - f)(\mathbf{G}) \, ds. \tag{7}$$

Furthermore, with the squared norm of the difference to make sure the direction of the curve does not affect the result, integrating over all pairs $(p, q)$, averaging, and considering the expectation under the distribution $p(\mathbf{G}|\mathbf{e} = e_i)$, we can reformulate the optimization target at test time as:

$$\arg\min_\psi \mathcal{L}_{\text{symm.}} = \int \left\| \int_{1 \leq p < q \leq |v|} \gamma_{p,q} \nabla(f \circ g_\psi - f)(\mathbf{G}) \, ds \right\|^2 p(\mathbf{G}|\mathbf{e} = e_i) \, d\mathbf{G}. \tag{8}$$

This formula captures the cumulative effect of the gradient along the paths between the augmented graphs, encouraging smoothness and consistency in the representations learned by the adapter $g_\psi$.

**Consistency.** To further enhance the robustness of the adaptation process to the nonlinear mappings of GNNs both in the input graph space and the embedding space, we design the adapter $g_\psi$ to be exchangeable with the pre-trained GNN $f$, i.e., $g_\psi(f(\mathcal{G})) \approx f(g_\psi(\mathcal{G}))$. This design encourages $g_\psi$ to learn an *isomorphic mapping* with $f$, ensuring that the transformations applied by the adapter are compatible with those of the GNN. The benefit of this approach is that it preserves the structural and semantic information of the graphs during adaptation, leading to more robust and consistent representations across different environments.

To formalize this idea, we consider the paths $\gamma_{p,p}$ connecting the embeddings obtained by $f(g_\psi(\mathcal{G}_p))$ and $g_\psi(f(\mathcal{G}_p))$ in the embedding space. According to the fundamental theorem of calculus, the difference between these two mappings can be expressed as an integral of the gradient along the path $\gamma_{p,p}$: [4]

$$f(g_\psi(\mathcal{G}_p)) - g_\psi(f(\mathcal{G}_p)) = \int \gamma_{p,p} \nabla(f \circ g_\psi - g_\psi \circ f)(\mathbf{G}) \, ds, \tag{9}$$

Same as $\mathcal{L}_{\text{symm.}}$, but cancels out the effects of subtraction order and the expectation in environments $e_i$ that produce OOD distributions, we can reformulate the optimization target as:

$$\arg\min_\psi \mathcal{L}_{\text{con.}} = \int \left\| \int_{1 \leq p \leq |v|} \gamma_{p,p} \nabla(f \circ g_\psi - g_\psi \circ f)(\mathbf{G}) \, ds \right\|^2 p(\mathbf{G}|\mathbf{e} = e_i) \, d\mathbf{G}, \tag{10}$$

which is subject to the constraint:

$$\int f(g_\psi(\mathcal{G}_p)) \, p(\mathbf{G}|\mathbf{e} = e_i) \, d\mathbf{G} < \epsilon. \tag{11}$$

Denote that $\epsilon$ is any number greater than 0. The left term of this restriction can be directly used as an optimization target $\mathcal{L}_{\text{R}}$. By enforcing the isomorphism between $f$ and $g_\psi$, we promote the preservation of structural information and ensure that the adapted representations remain meaningful within the context of the original model. This alignment leads to improved generalization and robustness when adapting to out-of-distribution environments. The overall optimization objective for our adapter's test-time tuning can be written in the following form: [5]

$$\arg\min_\psi \mathcal{L}_{\text{A2A}} = \alpha\lambda(\mathcal{L}_{\text{symm.}} + \mathcal{L}_{\text{con.}}) + (1 - \alpha)\mathcal{L}_{\text{R}}, \tag{12}$$

where $\alpha$ and $\lambda$ are hyperparameters that control the importance of each objective. $\alpha$ is a hyperparameter in the range $(0, 1)$; $\lambda$ is a positive hyperparameter.

---

[4] $(g_\psi \circ f)(\mathbf{G}) = g_\psi(f(\mathcal{G})) = \mathbf{H}_{\mathbf{X}} + \mathbf{H}_{\hat{\mathbf{E}}} = f(\mathbf{A}, \mathbf{X}) + f(\mathbf{A}, \hat{\mathbf{E}})$.

[5] A practical discrete form of $\mathcal{L}_{\text{A2A}}$ can be found in the Appendix.A.2 and $\mathcal{L}_{\text{A2A}}$ under two augmented graphs views, i.e. $|v| = 2$, can be found in the Appendix.A.1).

Table 2: Average classification performance (%) on the test graphs. The best performance on each dataset with a specific backbone is indicated in bold, the second-best method is underlined, and C. indicates the average ranking of the same method compared to others on all six datasets under the same backbone. OOM indicates an out-of-memory error on 24 GB GPU memory. $^{\uparrow}$/* indicates that GOAT outperforms ERM at the confidence level 0.1/0.05 from the paired t-test.

| Backbone | Method | Amz-Photo | Cora | Elliptic | FB-100 | OGB-Arxiv | Twitch-E | C. |
|---|---|---|---|---|---|---|---|---|
| GCN | ERM | 92.78±1.34 | 93.92±0.64 | 54.13±1.18 | 53.95±0.77 | 36.89±0.67 | 56.84±1.13 | 4.7 |
| | EERM | 94.24±0.40 | 87.36±0.86 | 53.15±0.01 | 54.03±0.80 | OOM | 57.25±0.42 | 5.2 |
| | Tent | 93.84±1.53 | 91.64±2.37 | 46.72±0.06 | 54.11±1.50 | 39.34±2.76 | 60.01±0.95 | 4.2 |
| | GCTA | 91.43±1.74 | 93.13±2.02 | 55.82±3.50 | 54.11±1.49 | 37.27±3.46 | 60.10±0.95 | 3.7 |
| | GTrans | 94.32±1.34 | 94.76±1.94 | 55.07±3.61 | 54.17±1.23 | **40.45±1.76** | **60.37±1.44** | 1.8 |
| | GOAT | **94.35±1.32$^{\uparrow}$** | **94.79±1.36** | 55.83±3.81 | 54.19±2.04 | 39.44±2.02* | 60.15±1.30* | **1.3** |
| SAGE | ERM | 87.79±1.74 | 99.62±0.09 | 50.11±0.39 | 54.09±0.40 | 37.52±0.66 | 59.20±0.14 | 5.2 |
| | EERM | 95.76±0.11 | 99.76±0.21 | 60.43±0.29 | OOM | OOM | 60.09±0.25 | 5.2 |
| | Tent | 95.23±1.52 | 99.71±0.17 | 50.25±3.28 | 55.11±0.55 | 39.56±1.49 | **62.05±0.22** | 3.0 |
| | GCTA | 96.86±1.11 | 99.85±0.06 | 66.92±2.33 | 55.11±0.56 | 33.67±3.25 | 62.05±0.24 | 3.2 |
| | GTrans | **97.09±1.13** | 99.81±0.16 | 63.04±6.39 | 55.07±0.59 | **39.74±1.14** | 61.97±0.34 | 2.5 |
| | GOAT | 92.54±2.51* | **99.89±0.10*** | **67.92±5.56*** | **55.61±0.30*** | 39.52±1.03* | 61.91±0.28* | **2.5** |
| GAT | ERM | 94.92±2.33 | 95.99±0.88 | 49.49±1.51 | 48.25±1.55 | 37.92±0.68 | 57.36±0.30 | 3.8 |
| | EERM | 94.07±1.32 | 79.35±8.90 | 54.27±2.42 | 52.46±2.02 | OOM | 56.27±0.37 | 3.7 |
| | Tent | 94.96±0.87 | 93.54±3.50 | 55.29±5.22 | 51.22±1.99 | 37.41±5.20 | 58.93±1.50 | 5.3 |
| | GCTA | 94.72±1.73 | **96.03±1.76** | 56.00±10.11 | 51.22±1.98 | 37.86±2.17 | 58.83±1.59 | 3.2 |
| | GTrans | **95.14±0.70** | 95.46±1.96 | **62.56±4.22** | 51.27±1.91 | 37.52±2.68 | 58.84±1.49 | 2.5 |
| | GOAT | 94.69±0.63 | 94.72±2.83 | 60.33±4.83* | **54.20±1.10*** | **41.13±1.96*** | **58.95±1.50*** | **2.3** |
| GPR | ERM | 84.81±3.71 | 83.98±1.72 | 48.96±1.05 | 54.36±0.27 | 40.91±0.28 | 57.25±0.66 | 4.1 |
| | EERM | 90.87±0.52 | 87.16±2.39 | 60.08±0.03 | 54.21±0.42 | OOM | 58.75±0.29 | 4.0 |
| | Tent* | - | - | - | - | - | - | - |
| | GCTA | 91.96±0.75 | 92.75±2.48 | 66.36±3.67 | 54.63±0.77 | 44.44±0.70 | 59.97±0.62 | 2.6 |
| | GTrans | 91.97±0.84 | 92.70±2.46 | **68.54±5.56** | 54.48±0.66 | **45.64±0.61** | 59.84±0.89 | 2.4 |
| | GOAT | **91.98±0.83*** | **92.79±2.74*** | 66.47±6.44* | **55.23±0.43*** | 44.78±0.69* | **60.00±0.65*** | **1.7** |

* Tent cannot be applied to models that do not contain batch normalization layers.

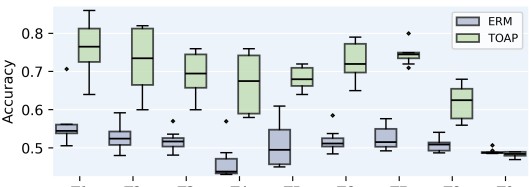

Figure 4: Results on Elliptic under OOD. GOAT improves SAGE on most test graphs.

| | Running Time (s) | | | GPU Memory (GB) | | |
|---|---|---|---|---|---|---|
| | Photo | Ellip. | ArXiv | Photo | Ellip. | ArXiv |
| EERM | 413.4 | 629.6 | - | 10.5 | 12.8 | 24+ |
| GTrans | 1.9 | 6.8 | 12.2 | 1.6 | 1.3 | 4.1 |
| GOAT | 5.5 | 0.5 | 0.3 | 1.5 | 1.3 | 5.0 |

Table 3: Efficiency comparison. GOAT is more time- and memory-efficient than EERM on large graphs and comparable to GTrans.

## 4 EXPERIMENTS

### 4.1 GENERALIZATION ON OUT-OF-DISTRIBUTION DATA

**Datasets:** The experiments validate GOAT on three types of distribution shifts using six benchmark datasets, following the settings in EERM (Wu et al., 2022) which is designed for node-level tasks on OOD data. These include (1) **artificial transformation** for Cora (Yang et al., 2016) and Amazon-Photo (Shchur et al., 2018), (2) **cross-domain** transfers for Twitch-E (Rozemberczki et al., 2021) and FB-100 (Traud et al., 2012), and (3) **temporal evolution** for Elliptic (Pareja et al., 2020) and OGB-ArXiv (Hu et al., 2020). The datasets are split into training/validation/test sets with ratios of 1/1/8 for Cora and Amazon-Photo, 1/1/5 for Twitch-E, 3/2/3 for FB-100, 5/5/33 for Elliptic, and 1/1/3 for OGB-ArXiv. More details on the datasets can be found in Appendix D.1.

**Baselines.** GOAT is compared with four baselines: empirical risk minimization ERM, test-time training method Tent (Wang et al.), memory-bank based method GraphCTA(GCTA) (Zhang et al., 2024), the train-time state-of-the-art method EERM (Wu et al., 2022) which is exclusively developed for graph OOD issues, and the test-time graph transformation state-of-the-art method GTrans (Jin

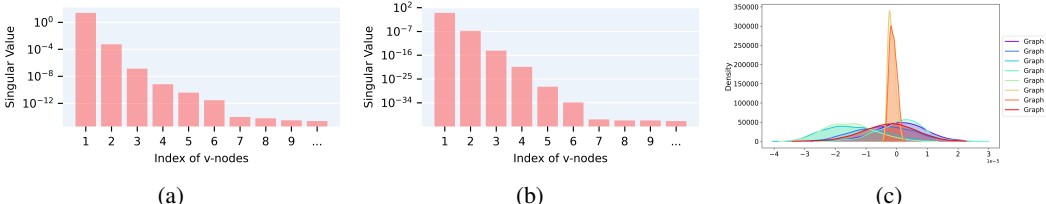

(a)                          (b)                          (c)

Figure 5: (a)(b) Visualization of the low-rank property of matrix $E$ in the LROG module of a GAT backbone trained on the two largest test graphs on OGB-ArXiv(169343 nodes) and FB-100(41554 nodes) under OOD settings. The singular values, obtained via SVD, show a rapid decay, indicating that **node embeddings can be effectively compressed into virtual nodes of the units digit**. (c) Visualization of the distribution of generated OOD representation obtained after training GOAT on 8 test graphs in Cora with Gaussian KDE. The x-axis represents the sum of feature values on the nodes, and the y-axis represents the density of bias on each node within that value range. The further the mode of the distribution is from "0", the greater the degree of OOD.

et al., 2022). All methods are evaluated with four popular GNN backbones: GCN (Kipf and Welling, 2016), GraphSAGE (Hamilton et al., 2017), GAT (Veličković et al., 2018), and GPR (Chien et al., 2020). Their default setup follows that in EERM[6]. More implementation details of the baselines and GOAT can be found in Appendix D.2. All experiments are repeated 8 times with different random seeds. Due to page limits, additional baselines and backbones such as SR-GNN (Zhu et al., 2021), UDA-GCN (Wu et al., 2020), and GTN (Yun et al., 2019) are included in Appendix E.

**Overall Comparison.** Table 2 reports the averaged performance over the test graphs for each dataset as well as the average rank of each algorithm. From the table, we conduct the following observations: (a) Overall Performance. The proposed framework consistently achieves strong performance across the datasets: GOAT achieves average ranks of 1.3, 2.5, 2.3, and 1.7 with GCN, SAGE, GAT, and GPR, respectively, while the corresponding ranks for the best baseline GOAT are 1.8, 2.5, 2.3 and 2.4. Furthermore, in most cases, GOAT significantly improves the vanilla baseline (ERM) by a large margin. Particularly, when using SAGE as the backbone, GOAT outperforms ERM by 9.8%, 18.5%, and 3.9% on Cora, Elliptic, and OGB-ArXiv, respectively. These results demonstrate the effectiveness of GOAT in tackling diverse types of distribution shifts. (b) Comparison to other baselines. Both GraphCTA and EERM modify the model parameters to improve model generalization. Nonetheless, they are less effective than GOAT, as GOAT takes advantage of adapting the pre-trained GNN to the environment of test graphs. As test-time training methods, Tent and GTrans also perform well in some cases. However, Tent is ineffective for models that do not incorporate batch normalization. On the other hand, GTrans not only modifies node features but also alters edges, which can backfire if the edge modifications are not carefully chosen, potentially leading to a misrepresentation of the graph structure.

We further show the performance on each test graph on Elliptic with SAGE in Figure 4 and the results for other datasets are provided in Appendix.Figure 8. We observe that GOAT generally improves over individual test graphs within each dataset, which validates the effectiveness of GOAT.

**Efficiency Comparison.** In Table 3, we compare the computational time and GPU usage on the largest graph of each dataset for our GOAT, EERM, and GTrans methods. Unlike EERM, which increases pre-training generalization through extensive environment augmentation during train time, GOAT optimizes efficiency by minimizing reliance on computationally expensive data augmentations and parameter tuning. In contrast to GTrans, which adjusts based on the proportion of edges modified on the graph, GOAT requires sampling only a minimal number of two OOD graph views per training epoch. These features ensure that GOAT not only conserves GPU resources but also accelerates the adaptation process during test time, showcasing substantial efficiency improvements over both train time and other test-time methods.[7]

---

[6]Adjustments have only been made to the hidden dimensions of GAT to ensure consistency in the parameter count across all four backbones.

[7]Detailed early-stop procedures are shown in Appendix C.

Table 4: Ablation study of the loss function $\mathcal{L}_{\text{A2A}}$ comparison on the Elliptic dataset under OOD. Two-view sampling under a test environment shows improvement in GCN average performance on test graphs with the addition of each $\mathcal{L}_{\text{A2A}}$ constraint component, demonstrating the effectiveness of each part of the loss function and the choice of the number of samples.

| (a) Two samples | | | | | (b) One sample | | | | |
| --- | --- | --- | --- | --- | --- | --- | --- | --- | --- |
| Pre-train | $\mathcal{L}_{\text{symm.}}$ | $\mathcal{L}_{\text{con.}}$ | $\mathcal{L}_{\mathbf{R}}$ | Performance | Pre-train | $\mathcal{L}_{\text{symm.}}$ | $\mathcal{L}_{\text{con.}}$ | $\mathcal{L}_{\mathbf{R}}$ | Performance |
| ✓ | | | | ±0.00% | ✓ | | | | ±0.00% |
| ✓ | ✓ | | | -1.51% | ✓ | ✓ | | | -1.51% |
| ✓ | ✓ | ✓ | | +4.49% | ✓ | ✓ | ✓ | | -1.62% |
| ✓ | ✓ | ✓ | ✓ | **+5.28%** | ✓ | ✓ | ✓ | ✓ | -0.18% |

## 4.2 Low-rank of Node-level Representation on Large Graph

After tuning the parameters of LROG on validation sets and obtaining the optimal results through test tuning, we further investigate the principal components of the low-rank matrix $\mathbf{E}$ in the $N$-dimensional space as Figure 5(a)(b) shown. By performing Singular Value Decomposition (SVD), we obtained the singular values sorted in descending order and compared the major eigenvalues that showed a significant decline compared to the others. In almost all large graphs, the dimensions of OOD representation we obtained were low-rank. This is different from the low-rank attention on the node feature level for each node's dimensions. This further demonstrates that, in the new environment where OOD graphs are generated, the environment can be generalized by a low-rank addable representation. This also provides empirical evidence for setting our $|n|$ hyperparameter to a small constant that is independent of the number of nodes.

## 4.3 Quantifying Distribution Shift

We utilize Kernel Density Estimation (KDE) to visualize the distribution of OOD representation generated by our adapter obtained through the GOAT method on OOD datasets as Figure 5(c) shown, by aggregating each node's feature dimensions $d$. As the initialization of the OOD representation is zero, the mean and mode of the initial distribution should be 0. Due to the varying degrees of OOD in different graphs, after tuning by GOAT, our adapter can effectively capture the discrepancy between the current test graph and the pre-trained GNN. Adding the generated OOD representation can be seen as the mapping from the current test graph to the distribution to which the original training graph belongs. Therefore, the farther the representation deviates from the origin, the more severe distribution shift the graph has, whether observed from the perspective of the entire graph or an individual node's perspective. The distribution shifts in the time-evolving graph could be more intuitive as time flows, we show the other OOD representation's distribution in other datasets and compare with central moment discrepancy (CMD) (Zellinger et al., 2017) measurement in Appendix.Figure 9, highlighting the interpretability of our designed adapter.

## 4.4 Ablation Studies and Parameter Study

**Optimization Target $\mathcal{L}_{\text{A2A}}$.** Ablation study shown in Table 4 demonstrates the effectiveness of the different components in our proposed loss function. Optimizing $\mathcal{L}_{\text{symm.}}$ alone may lead to instability and mode collapse. To fully leverage the point estimation effect of $\mathcal{L}_{\text{symm.}}$, it is essential to satisfy the constraints imposed by $\mathcal{L}_{\mathbf{R}}$ and $\mathcal{L}_{\text{con.}}$. $\mathcal{L}_{\mathbf{R}}$ ensures that the OOD can be represented by the GNN and serve as an addable tensor effect in the representation space, while $\mathcal{L}_{\text{con.}}$ acts as a self-supervised loss to preserve the equal effect on input and representation space.

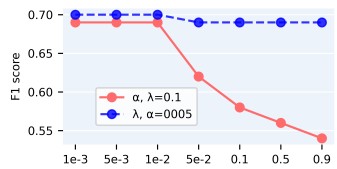

Figure 6: $\alpha, \lambda$ Parameter Study

The best performance is achieved when all three components are jointly optimized. Moreover, at least two views should be sampled so that the learned information isn't biased. This underscores the importance of the constraints in Eq.(11)

**Adapter LROG.** In Figure 6, we show the parameter study of $\lambda$ and $\alpha$ in $\mathcal{L}_{\text{A2A}}$. Noted that there could be a different proportion of $\frac{\lambda}{(1-\alpha)}$, while it still should be a relatively larger value of $\alpha$ in that the constraint in Eq.(11), i.e. $\mathcal{L}_{\mathbf{R}}$, should be satisfied first then the other objective could work.

### 4.5 Further Analysis

**Universal Bias vs. Local-global Bias.** We compare the average improvement of various non-customized additional parameter methods using our proposed $\mathcal{L}_{\text{A2A}}$ on OOD datasets in test time. For node classification, it is evident that UPF's global prompts (Fang et al., 2024), $\boldsymbol{E} \in \mathbb{R}^{1 \times d}$, across all nodes, are less customizable for classifying each node in an OOD environment, and might even learn controversial knowledge. Moreover, using a selection dictionary (Sun et al., 2023), $\boldsymbol{E} \in \mathbb{R}^{k \times d}$ ($k \ll N$), also presents difficul-

Table 5: Bias Comparison

| Method | Avg. Impr |
|--------|-----------|
| Universal | +0.01% |
| Prompt dict | +0.01% |
| Sub-graph | +1.52% |
| Node-wise | +2.35% |

ties during test-time training. In contrast, subgraph-focused methods (Lee et al., 2024; Sun et al., 2022) can simultaneously capture the optimal bias more effectively, yielding relatively higher results, especially when it extends to a node-wise bias, i.e. each node's learnable bias is different, $\boldsymbol{E} \in \mathbb{R}^{N \times d}$. These demonstrate that at least in OOD node classification, bias design that focuses on local-global context can better capture the relationships of nodes within the OOD environment. The node-wise bias method is particularly well-suited to our augmentation-to-augmentation strategy, as it can better adapt to OOD scenarios. This further validates the rationality of our adapter's design. Furthermore, based on the methods used by Liu et al. (2023b) and Yu et al. (2023), we experimented with incorporating a learnable scaling parameter that multiplies the weights of each GNN layer or node embeddings during test time. However, we found this approach difficult to apply effectively in our context.

## 5 Conclusion

We propose a novel augmentation-to-augmentation approach to effectively adapt pre-trained GNNs to unlabeled OOD test graphs, regardless of the pre-training architecture or method. We introduce a new $\mathcal{L}_{\text{A2A}}$ self-supervised loss function, which enhances the extraction of inductive information from pre-trained GNNs. Additionally, by employing a low-rank adapter to the node feature, the inductive generation of OOD representation becomes more efficient. Our method, GOAT, generally outperforms state-of-the-art techniques across all datasets. An intriguing future direction is to extend such adapter to other tasks involving OOD distributions (such as graph classification and edge prediction) and delve into its availability to model OOD distribution that generates discrete edges.

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

# A    PROOFS & EXAMPLES

**Proposition a.  (Which has been proved in GPF(Fang et al., 2024))** Given a pre-trained GNN model $f$, an input graph $\mathcal{G}$, for any graph-level transformation $g : \mathcal{G}\{\boldsymbol{A}, \boldsymbol{X}\} \rightarrow \mathcal{G}'\{\boldsymbol{A}', \boldsymbol{X}'\}$, there exists an additional extra feature vector $\hat{\boldsymbol{E}}$ that satisfies:

$$f(\boldsymbol{A}, \boldsymbol{X} + \hat{\boldsymbol{E}}) = f(g(\boldsymbol{A}, \boldsymbol{X})) = f(\boldsymbol{A}', \boldsymbol{X}')$$

## A.1    AN INTUITIVE EXAMPLE UNDER TWO VIEWS SAMPLED FROM TEST TIME ENVIRONMENT AND RATIONALE

Assuming that there is a test Graph: $\mathcal{G}_{te} = \{\boldsymbol{A}', \boldsymbol{X}'\}$, and two different views of $\mathcal{G}_{te}$, denoted by $\mathcal{G}_1$ and $\mathcal{G}_2$, a pre-trained GNN model $f$, then we have:

$$\mathcal{G}_1 = g_1(\mathcal{G}_{te}) = (\boldsymbol{A}_1', \boldsymbol{X}_1') \tag{Ap.1}$$

$$\mathcal{G}_2 = g_2(\mathcal{G}_{te}) = (\boldsymbol{A}_2', \boldsymbol{X}_2') \tag{Ap.2}$$

According to **Proposition a**, there separately exist two respective representation: $\hat{\boldsymbol{E}}_1$ and $\hat{\boldsymbol{E}}_2$, making the following formula true:

$$f(\boldsymbol{A}_1', \boldsymbol{X}_1') = f(\boldsymbol{A}', \boldsymbol{X}' + \hat{\boldsymbol{E}}_1) \tag{Ap.3}$$

$$f(\boldsymbol{A}_2', \boldsymbol{X}_2') = f(\boldsymbol{A}', \boldsymbol{X}' + \hat{\boldsymbol{E}}_2) \tag{Ap.4}$$

Let $\mathcal{G}'^* = \{\boldsymbol{A}'^*, \boldsymbol{X}'^*\}$ represents the input graph at which the loss function $\mathcal{L}(f(\mathcal{G}^*), \boldsymbol{Y})$ is optimal.
**Proposition b.**  For $\forall(\boldsymbol{A}'', \boldsymbol{X}'')$, there exists a $\boldsymbol{E}'^*$ such that $f(\boldsymbol{A}'', \boldsymbol{X}'' + \boldsymbol{E}'^*) = f(\boldsymbol{A}'^*, \boldsymbol{X}'^*)$.

Naturally, it is desirable to design a loss function so that an augmented view $\mathcal{G}'$ is close enough to the $f$-mapped representation in the representation space of the $f$ mapped solution $\mathcal{G}'^*$:

$$P_{A2S} = \arg\min_{\hat{\boldsymbol{E}}_1} \mathbb{E}\left[\|f(\boldsymbol{A}_1', \boldsymbol{X}_1' + \hat{\boldsymbol{E}}_1) - f(\boldsymbol{A}'^*, \boldsymbol{X}'^*)\|^2\right]$$

$$= \arg\min_{\hat{\boldsymbol{E}}_1} \mathbb{E}\left[\|f(\boldsymbol{A}_1', \boldsymbol{X}_1' + \hat{\boldsymbol{E}}_1)\|^2 - 2f(\boldsymbol{A}'^*, \boldsymbol{X}'^*)^T f(\boldsymbol{A}_1', \boldsymbol{X}_1' + \hat{\boldsymbol{E}}_1)\right]. \tag{Ap.5}$$

While test-time tuning is a zero-shot task and we do not have prior knowledge about $\mathcal{G}^*$, instead we focus on the self-supervised views generated from the test graph $\mathcal{G}_{te}$ and make it equivalent to the supervised MSE loss fitting described above. Then we define the naive Augmentation-to-Augmentation target $P_{A2A}$ as follows:

$$P_{A2A} = \arg\min_{\hat{\boldsymbol{E}}} \mathbb{E}\left[\|f(\boldsymbol{A}_1', \boldsymbol{X}_1' + \hat{\boldsymbol{E}}_1) - f(\boldsymbol{A}_2', \boldsymbol{X}_2')\|^2\right]$$

$$= \arg\min_{\hat{\boldsymbol{E}}} \mathbb{E}\left[\|f(\boldsymbol{A}_1', \boldsymbol{X}_1' + \hat{\boldsymbol{E}}_1)\|^2 - 2f(\boldsymbol{A}_2', \boldsymbol{X}_2')^T f(\boldsymbol{A}_1', \boldsymbol{X}_1' + \hat{\boldsymbol{E}}_1)\right]. \tag{Ap.6}$$

According to **Proposition b**, that $f(\boldsymbol{A}_2', \boldsymbol{X}_2' + \boldsymbol{E}_2'^*) = f(\boldsymbol{A}'^*, \boldsymbol{X}'^*)$, Ap.5 is equivalent to:

$$\arg\min_{\hat{\boldsymbol{E}}_1} \mathbb{E}\left[\|f(\boldsymbol{A}_1', \boldsymbol{X}_1' + \hat{\boldsymbol{E}}_1)\|^2 - 2f((\boldsymbol{A}_2', \boldsymbol{X}_2' + \boldsymbol{E}_2'^*)^T f(\boldsymbol{A}_1', \boldsymbol{X}_1' + \hat{\boldsymbol{E}}_1)\right]. \tag{Ap.7}$$

However, there still exists a gap between Ap.6 and Ap.7 which represent $P_{A2S}$ and $P_{A2A}$ respectively. To make the two equivalent, it is necessary to satisfy:

$$\mathbb{E}\left[f(\boldsymbol{A}_2', \boldsymbol{X}_2' + \boldsymbol{E}_2^*) - f(\boldsymbol{A}_2', \boldsymbol{X}_2')\right] = 0. \tag{Ap.8}$$

Optimizing Eq.Ap.8 is equivalent to allowing for letting $f(\boldsymbol{A}'', \boldsymbol{X}'' + \hat{\boldsymbol{E}}') - f(\boldsymbol{A}'', \boldsymbol{X}'') = 0$ or itself $\mathbb{E}\left[f(\boldsymbol{A}'', \boldsymbol{X}'' + \hat{\boldsymbol{E}}') - f(\boldsymbol{A}'', \boldsymbol{X}'')\right] = 0$, it often leads to a collapse mapping so that all the augmented graphs in the representation space would be mapped to the same point. This makes it impossible to mitigate the distribution shift of the input test graph $\mathcal{G}_{te}$ in the representation space. So we relax the conditions with  but add constraints to avoid the collapse problem, and we get the following equivalent optimization goals defined:

$$\mathbb{E}\left[\|f(\boldsymbol{A}_2', \boldsymbol{X}_2' + \boldsymbol{E}_2^*) - f(\boldsymbol{A}_2', \boldsymbol{X}_2') - f(\boldsymbol{A}_2', \boldsymbol{E}_2^*)\|_p^p\right] = 0, \tag{Ap.9}$$

i.e.$(p = 2)$

$$\arg\min_{\hat{E}_2} \mathbb{E}\|f(\mathbf{A'_2}, \mathbf{X'_2} + \hat{E}_2) - f(\mathbf{A'_2}, \mathbf{X'_2}) - f(\mathbf{A'_2}, \hat{E}_2))\|^2,$$

$$\text{s.t.} \quad \mathbb{E}(f(\mathbf{A'_2}, \hat{E}_2)) = 0.$$

Thus, in our experiment, the full losses function we use are defined as follows:

$$\mathcal{L}_{\text{symm.}} = \frac{1}{2}\left(\left\|f(\boldsymbol{A'_1}, \boldsymbol{X'_1} + \hat{E}_1) - f(\boldsymbol{A'_2}, \boldsymbol{X'_2})\right\|^2 + \left\|f(\boldsymbol{A'_2}, \boldsymbol{X'_2} + \hat{E}_2) - f(\boldsymbol{A'_1}, \boldsymbol{X'_1})\right\|^2\right),$$

$$\mathcal{L}_{\text{con.}} = \frac{1}{2}\left(\left\|f(\boldsymbol{A'_1}, \boldsymbol{X'_1} + \hat{E}_1) - f(\boldsymbol{A'_1}, \boldsymbol{X'_1}) - f(\mathbf{A_1}, \hat{E}_1)\right\|^2 + \left\|f(\boldsymbol{A'_2}, \boldsymbol{X'_2} + \hat{E}_2) - f(\boldsymbol{A'_2}, \boldsymbol{X'_2}) - f(\boldsymbol{A'_2}, \hat{E}_2)\right\|^2\right),$$

$$\mathcal{L}_{\text{R}}(\hat{E}) = \mathbb{E}[f(\mathbf{A_1}, \hat{E}_1) + f(\mathbf{A_2}, \hat{E}_2)].$$

## A.2 DISCRETE FORM OF $\mathcal{L}_{\text{A2A}}$

$$\mathcal{L}_{\text{symm.}} = \mathbb{E}_{\substack{\mathcal{G}_{p,q} \sim p(\mathbf{G}|\mathbf{e}=\boldsymbol{e}_i) \\ 1 \le p < q \le |v|}}\left[\frac{1}{\binom{|v|}{2}} \sum_{|v|}\sum_{|v|}\|(f(g_\psi(\mathcal{G}_p)), f(g_\psi(\mathcal{G}_q)))\|^2\right],$$

$$\mathcal{L}_{\text{con.}} = \mathbb{E}_{\substack{\mathcal{G}_p \sim p(\mathbf{G}|\mathbf{e}=\boldsymbol{e}_i) \\ 1 \le p \le |v|}}\left[\frac{1}{|v|} \sum_{|v|}\|(f(g_\psi(\mathcal{G}_p)), g_\psi(f(\mathcal{G}_p)))\|^2\right],$$

$$\mathcal{L}_{\text{R}} = \mathbb{E}_{\substack{\mathcal{G}_p \sim p(\mathbf{G}|\mathbf{e}=\boldsymbol{e}_i) \\ 1 \le p \le |v|}}[f(g_\psi(\mathcal{G}_p))],$$

where $\mathcal{G}_{p,q} \sim p(\mathbf{G}|e = \boldsymbol{e}_i)$ denote as the augmented views sampled from the test OOD environment $\boldsymbol{e}_i$. $|v|$ is the number of augmented views.

## A.3 DISCUSSION

Furthermore, this also indicates that OOD representation should be generated for each specific test graph if the generation of OOD representation is an adapter that could learn a certain distribution, we can better form this idea into a point estimation.

Also, if only using a learnable parameter with the same size as $\boldsymbol{X'}$ that is not correlated with the test graph, there could reach a sub-optimal solution, but also work. In **Proposition b.** "$a \ \hat{\boldsymbol{E}}^{*'}$" should be turn into "$a$ common OOD representation $\hat{\boldsymbol{E}}^*$".

## B  A FIGURE ILLUSTRATION OF THE GOAT PARADIGM

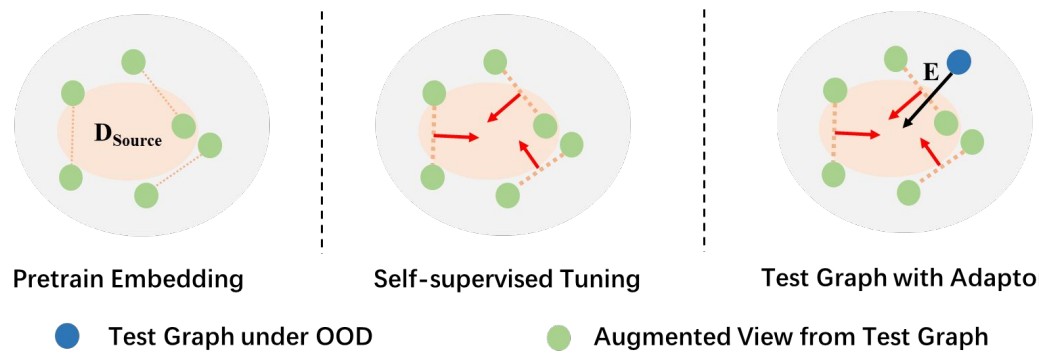

Figure 7: An illustration of **GOAT** with the proposed Loss $\mathcal{L}_{\textbf{A2A}}$, in two smapled views method
.

Assume we pre-train a GNN on train-time dataset ($D_{\text{Source}}$, represented by the orange section in the figure) to avoid overconfidence and overfitting. During test time, the non-IID test graph may map outside the original $D_{\text{Source}}$ (possibly with or without intersection), resulting in poor performance in

the GNN's decision space. However, by sampling different views in each gradient descent iteration, our loss function trains the OOD representation added to the nodes in the test graph. As shown in Figure 7, this effectively fits the normal of a hyperplane shared by the sampled views and includes a gradient direction towards the expected center of all views. Consequently, the mapping of the final test graph is closer to the decision space that the pre-trained GNN has learned to make correct decisions.

## C  ALGORITHM

---

**Algorithm 1 GOAT** for Test-Time OOD Graph Adaptation

---

1: **Input:** Pre-trained GNN $f_{\theta^*}$ ($\theta^*$ is fixed, without the last layer which is the classifier head) and test graph $\mathcal{G}_{te} = \{\boldsymbol{A}', \boldsymbol{X}'\}$, Sample method(DropEdge) $\mathcal{A}(\cdot)$
2: **Output:** Model prediction $\hat{Y}$ and OOD representation $\hat{\boldsymbol{E}}$
3: Initialize LROG$_\psi, \alpha, \lambda$
4: $\mathcal{L}_{\text{best}} = \infty$, patience = k, patience$_{\text{now}} = 0$
5: **for** $t = 1$ to $T$ **do**
6:    $\mathcal{G}' = \mathcal{A}(\mathcal{G}_{te}) = \{\mathbf{A}', \mathbf{X}'\}, \mathcal{G}'' = \mathcal{A}(\mathcal{G}_{te}) = \{\mathbf{A}'', \mathbf{X}''\}$
7:    $\hat{\boldsymbol{E}}' = \text{LROG}_\psi(\boldsymbol{A}', \boldsymbol{X}'), \hat{\boldsymbol{E}}'' = \text{LROG}_\psi(\boldsymbol{A}'', \boldsymbol{X}'')$
8:    $\hat{\boldsymbol{E}}'_{emb} = f_{\theta^*}(\mathbf{A}', \hat{\boldsymbol{E}}'), \hat{\boldsymbol{E}}''_{emb} = f_{\theta^*}(\mathbf{A}'', \hat{\boldsymbol{E}}'')$
9:    $\boldsymbol{H}'_{p_{emb}} = f_{\theta^*}(\mathbf{A}', \mathbf{X}'+\hat{\boldsymbol{E}}'), \boldsymbol{H}''_{p_{emb}} = f_{\theta^*}(\mathbf{A}'', \mathbf{X}''+\hat{\boldsymbol{E}}'')$
10:   $\boldsymbol{H}'_{emb} = f_{\theta^*}(\boldsymbol{A}',\mathbf{X}'), \boldsymbol{H}''_{emb} = f_{\theta^*}(\boldsymbol{A}'', \mathbf{X}'')$
11:   $\mathcal{L}_{\text{symm.}} = \mathbb{E}\|(\boldsymbol{H}'_{p_{emb}} - \boldsymbol{H}''_{emb}) + (\boldsymbol{H}''_{p_{emb}} - \boldsymbol{H}'_{emb})\|^2$
12:   $\mathcal{L}_{\text{con.}} = \mathbb{E}\|(\boldsymbol{H}'_{p_{emb}} - \boldsymbol{H}'_{emb} - \hat{\boldsymbol{E}}'_{emb}) + (\boldsymbol{H}''_{p_{emb}} - \boldsymbol{H}''_{emb} - \hat{\boldsymbol{E}}''_{emb})\|^2$
13:   $\mathcal{L}_{\text{R}} = \mathbb{E}|\hat{\boldsymbol{E}}'_{emb} + \hat{\boldsymbol{E}}''_{emb}|$
14:   $\mathcal{L} = \alpha\lambda(\mathcal{L}_{\text{symm.}} + \mathcal{L}_{\text{con.}}) + (1 - \lambda)\mathcal{L}_{\text{R}}$
15:   **Update:** $\psi \leftarrow \psi - \eta\Delta_\psi L$
16:   **if** $\mathcal{L}_{\text{R}} \leq \mathcal{L}_{best}$ **then**
17:       $\mathcal{L}_{best} = \mathcal{L}_{\text{R}}$
18:       patience$_{\text{now}} = 0$
19:   **else**
20:       patience$_{\text{now}} = $ patience$_{\text{now}} + 1$
21:   **if** patience$_{\text{now}} \geq$ patience **then**
22:       **Stop**
23: $\hat{\boldsymbol{E}} = \text{LROG}_\psi(\boldsymbol{A}', \boldsymbol{X}')$
24: $\hat{Y} = f_{\theta^*}(\boldsymbol{A}', \boldsymbol{X}' + \hat{\boldsymbol{E}})$
25: **return** $\hat{Y}$

---

## D  DATASETS AND HYPER-PARAMETERS

In this section, we reveal the details of reproducing the results in the experiments. We will release the source code upon acceptance.

### D.1  OUT-OF-DISTRIBUTION SETTING

The out-of-distribution (OOD) problem indicates that the model does not generalize well to the test data due to the distribution gap between training data and test data (Yang et al., 2021), which is also referred to as distribution shifts. Numerous research studies have been conducted to explore this problem and propose potential solutions (Zhu et al., 2021; Jin et al., 2022; Wu et al., 2022; Arjovsky et al., 2020; Ganin et al., 2016b; Muandet et al., 2013; Li et al., 2022a; Song and Wang, 2022). In the following, we introduce the datasets used for evaluating the methods that tackle the OOD issue in the graph domain.

**Dataset Statistics.** For the evaluation of OOD data, we use the datasets provided by EERM (Wu et al., 2022). The dataset statistics are shown in Table 6, which includes three distinct types of

Table 6: Summary of the experimental datasets that entail diverse distribution shifts.

| Dataset | Distribution Shift | #Nodes | #Edges | #Classes | Train/Val/Test Split | Metric | Adapted From |
|---|---|---|---|---|---|---|---|
| Cora | Artificial Transformation | 2,703 | 5,278 | 10 | Domain-Level | Accuracy | (Yang et al., 2016) |
| Amazon-Photo | | 7,650 | 119,081 | 10 | Domain-Level | Accuracy | (Shchur et al., 2018) |
| Twitch-explicit | Cross-Domain Transfers | 1,912 - 9,498 | 31,299 - 153,138 | 2 | Domain-Level | ROC-AUC | (Rozemberczki et al., 2021) |
| Facebook-100 | | 769 - 41,536 | 16,656 - 1,590,655 | 2 | Domain-Level | Accuracy | (Traud et al., 2012) |
| Elliptic | Temporal Evolution | 203,769 | 234,355 | 2 | Time-Aware | F1 Score | (Pareja et al., 2020) |
| OGB-ArXiv | | 169,343 | 1,166,243 | 40 | Time-Aware | Accuracy | (Hu et al., 2020) |

distribution shifts: (1) "Artificial Transformation" which indicates the node features are replaced by synthetic spurious features; (2) "Cross-Domain Transfers" transfers which means that graphs in the dataset are from different domains and (3) "Temporal Evolution" where the dataset is a dynamic one with evolving nature. Notably, we use the datasets provided by EERM(Wu et al., 2022), which were adopted from the aforementioned references with manually created distribution shifts. Note that there can be multiple training/validation/test graphs. Specifically, Cora and Amazon-Photo have 1/1/8 graphs for training/validation/test sets. Similarly, the splits are 1/1/5 on Twitch-E, 3/2/3 on FB-100, 5/5/33 on Elliptic, and 1/1/3 on OGB-ArXiv.

To show the performance on individual test graphs, we choose SAGE as the backbone model and include the box plot on all test graphs within each dataset in Figure 8. We observe that GOAT generally improves ERM over each test graph within each dataset, which validates the effectiveness of GOAT.

## D.2 HYPER-PARAMETER SETTING

For the setup of backbone GNNs, we majorly followed EERM (Wu et al., 2022):

(a) **GCN**: the architecture setup is 5 layers with 32 hidden units for Elliptic and OGB-ArXiv, and 2 layers with 32 hidden units for other datasets, with batch normalization for all datasets. The pre-train learning rate is set to 0.001 for Cora and Amz-Photo, 0.01 for other datasets; the weight decay is set to 0 for Elliptic and OGB-ArXiv, and 0.001 for other datasets.

(b) **GraphSAGE**: the architecture setup is 5 layers with 32 hidden units for Elliptic and OGB-ArXiv, and 2 layers with 32 hidden units for other datasets, and with batch normalization for all datasets. The pre-train learning rate is set to 0.01 for all datasets; the weight decay is set to 0 for Elliptic and OGB-ArXiv, and 0.001 for other datasets.

(c) **GAT**: the architecture setup is 5 layers for Elliptic and OGB-ArXiv, and 2 layers for other datasets, with batch normalization for all datasets. Each layer contains 4 attention heads and each head is associated with 8 hidden units. The pre-train learning rate is set to 0.01 for all datasets; the weight decay is set to 0 for Elliptic and OGB-ArXiv, and 0.001 for other datasets.

(d) **GPR**: We use 10 propagation layers and 2 transformation layers with 32 hidden units. The pre-train learning rate is set to 0.01 for all datasets; the weight decay is set to 0 for Elliptic and OGB-ArXiv, and 0.001 for other datasets. Note that **GPR does not contain batch normalization layers**.

For the baseline methods, we tuned their hyper-parameters based on the validation performance. For Tent, we search the learning rate in the range of $[1e-2, 1e-3, 1e-4, 1e-5]$ and the running epochs in $[1, 10, 20, 30]$. For EERM(Wu et al., 2022) and GTrans(Jin et al., 2022), we followed the instructions provided by the original paper. For GraphCTA(GCTA)(Zhang et al., 2024), we tune the feature adaptation $\eta_1$ in [5e-3, 1e-3, 1e-4, 1e-5, 1e-6], learning rate of structure adaptation $\eta_2$ in [0.5, 0.1, 0.01], and alternatively optimize node features epochs $\tau_1$ in [1, 2, 3] and optimize graph structure epochs $\tau_2$ in [1, 2], other parameters followed the instruction provided by the original paper. For GOAT, we adopt DropEdge as the augmentation function $\mathcal{A}(\cdot)$ and set the drop ratio to 0.05, $K$-layer aggregation in LROG set to 1 due to some GNN only has two layers in some datasets while the last GNN layer performs as a classifier head. We use Adam optimizer for LROG module tuning. We further search the learning rate $\eta$ in [1e-2, 5e-3, 1e-3, 5e-4, 1e-4, 5e-5, 1e-5, 1e-6] for different backbones, the virtual nodes number $|n|$ in $[1\times, 2\times, 5\times, 10\times, 20\times]$ of the class number

$C$, the attention dim $d_{attn}$ in LROG in [2, 4, 8, 16, 32], total epochs $T$ in [50, 100], and the patience in [1, 0.5, 0.1, 5e-2, 1e-2, 1e-3]. In the optimization target, we search the $\lambda$ in [1, 3, 5, 10] and the $\alpha$ in [0.999, 0.9, 0.75, 0.5, 0.25, 0.1, 5e-2, 1e-2, 5e-3]. We note that the process of tuning hyper-parameters is quick due to the high efficiency of test-time adaptation as we demonstrated in Section 4.1. Furthermore, not every test graph is learned over whole epochs set due to the patience of dissatisfaction of constraint in Eq.(10).

**Evaluation Protocol.** For ERM (standard pre-training), we pre-train all the GNN backbones using the common cross-entropy loss. For EERM, it optimizes a bi-level problem to obtain a trained classifier. Note that the aforementioned two methods do not perform any test-time adaptation and their model parameters are fixed during the test. For the four test-time adaptation methods, Tent, GCTA, GTrans, and GOAT. We first obtain the GNN backbones pre-trained from ERM and adapt the model parameters or graph data at test time, respectively. Furthermore, Tent minimizes the entropy loss and GTrans and GCTA both minimize the contrastive surrogate loss, while GOAT minimizes the Target $\mathcal{L}_{A2A}$.

### D.3 HARDWARE AND SOFTWARE CONFIGURATIONS

We perform experiments on NVIDIA GeForce RTX 3090 GPUs. The GPU memory and running time reported in Table 3 are measured on one single RTX 3090 GPU. Additionally, we use eight CPUs, with the model name as Intel(R) Xeon(R) Silver 4210R CPU @ 2.40GHz. The operating system utilized in our experiments was Ubuntu 22.04.3 LTS (codename jammy).

## E MORE EXPERIMENTAL RESULTS

### E.1 OVERALL COMPARISON

To show the performance on individual test graphs, we choose SAGE as the backbone model and include the box plot on all test graphs within each dataset in Figure 8. We observe that GOAT generally improves over each test graph within each dataset, which validates the effectiveness of our proposed method.

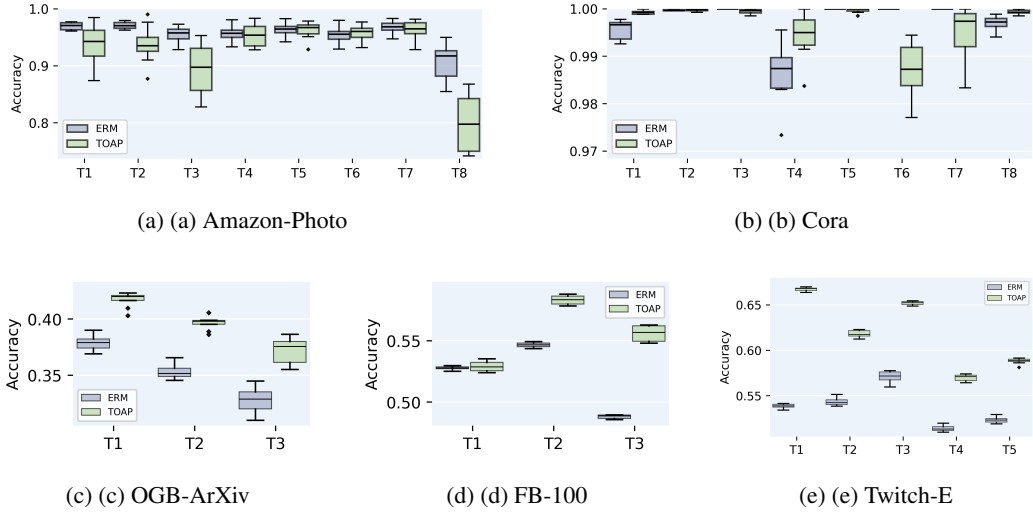

Figure 8: Classification performance on individual test graphs within each dataset for OOD setting.

### E.2 COMPARISON TO MORE BASELINE AND BACKBONES

To compare their empirical performance, we include two GraphDA methods (Zhu et al., 2021; Wu et al., 2020) and one general domain adaptation method (Ganin et al., 2016a). SR-GNN regularizes the model's performance on the source and target domains. Note that SR-GNN was originally

developed under the transductive setting where the training graph and test graph are the same. To apply SR-GNN in our OOD setting, we assume the test graph is available during the training stage of SR-GNN, as typically done in domain adaptation methods. UDA-GCN is another work that tackles graph data domain adaptation, which exploits local and global information for different domains. In addition, we also include DANN, which adopts an adversarial domain classifier to promote the similarity of feature distributions between different domains. We followed the authors' suggestions in their paper to tune the hyper-parameters and the results are shown in Table 7. On the one hand, we can observe that these graph domain adaptation methods generally improve the performance of GCN under distribution shift and SRGNN is the best-performing baseline. On the other hand, GOAT performs the best on all datasets except Amz-Photo. On Amz-Photo, GOAT does not improve as much as SR-GNN, which indicates that joint optimization over source and target is necessary for this dataset. However, recall that domain adaptation methods are less efficient due to the joint optimization on source and target. Overall, the test-time graph adaptation with our adapter could better fit the specific distribution shifts that deviate from the source target. As shown in Table 8, GOAT could also adapt to more popular backbones.

Table 7: Performance comparison between GOAT with GCN and graph domain adaptation methods.

| Method | Amz-Photo | Cora | Elliptic | FB-100 | OGB-ArXiv | Twitch-E |
|---|---|---|---|---|---|---|
| ERM | 93.79±0.97 | 91.59±1.44 | 50.90±1.51 | 54.04±0.94 | 38.59±1.35 | 59.89±0.50 |
| UDA-GCN | 91.70±0.35 | 92.65±0.46 | 51.57±1.31 | 54.11±0.54 | 39.43±0.71 | 52.12±0.38 |
| DANN | 94.08±0.21 | 92.89±0.64 | 53.00±0.97 | 51.53±1.47 | 36.60±1.26 | 60.13±0.53 |
| SRGNN | **94.64±0.17** | 94.08±0.28 | 51.94±0.81 | 54.08±1.10 | 38.92±0.65 | 59.21±0.51 |
| GOAT | 94.35±1.32 | **94.79±1.36** | **55.83±3.81** | **54.19±2.04** | **39.44±2.02** | **60.15±1.30** |

Table 8: Performance comparison between GOAT with other backbones.

| Method | Amz-Photo | Cora | Elliptic | FB-100 | OGB-ArXiv | Twitch-E |
|---|---|---|---|---|---|---|
| GTN | 94.73±2.91 | 99.88±0.10 | 68.51±3.85 | 53.57±0.75 | 43.08±0.84 | 62.30±0.16 |
| GTN + GOAT | 94.75±2.97 | 99.85±0.12 | 70.08±2.50 | 54.94±0.61 | 44.11±0.84 | 63.79±0.27 |

### E.3 QUANTIFYING DISTRIBUTION SHIFT THROUGH LROG

In this section, we further show more OOD representation distribution generated by GOAT that indicates the OOD degree of each test graph. We can see an extreme shift in Figure 9(c) that as the snapshots flow from the validation, the mode and the mean value of the OOD representation shift away from the 0 initialized value, which shows a further deviation of the later test graphs from the train source distribution. Furthermore, following SR-GNN (Zhu et al., 2021), we adopt central moment discrepancy (CMD) (Zellinger et al., 2017) as the measurement to quantify the distribution shifts in different graphs, we present them in Table 9 as a comparison with our OOD representation in GOAT.

Table 9: CMD values on each individual graph based on the pre-trained GCN.

| GraphID | $G_0$ | $G_1$ | $G_2$ | $G_3$ | $G_4$ | $G_5$ | $G_6$ | $G_7$ | $G_8$ |
|---|---|---|---|---|---|---|---|---|---|
| Amz-Photo | 6.4 | 5.1 | 5.5 | 3.7 | 2.8 | 3.7 | 3.9 | 6.6 | - |
| Cora | 5.4 | 4.2 | 4.8 | 6.3 | 5.5 | 4.8 | 4.6 | 5.4 | - |
| Elliptic | 80.2 | 90.8 | 114.3 | 86.5 | 789.3 | 781.6 | 99.4 | 100.4 | 150.6 |
| OGB-ArXiv | 14.7 | 20.6 | 10.4 | - | - | - | - | - | - |
| FB-100 | 29.7 | 16.9 | 32.9 | - | - | - | - | - | - |
| Twitch-E | 8.6 | 6.1 | 9.0 | 8.4 | 9.7 | - | - | - | - |

### E.4 LOW-RANK OF NODE-LEVEL REPRESENTATION ON LARGE GRAPH

In Figure 10, we show other E in LROG OOD representation generation on two large test graphs in OGB-arXiv with 69499 and 120740 nodes.

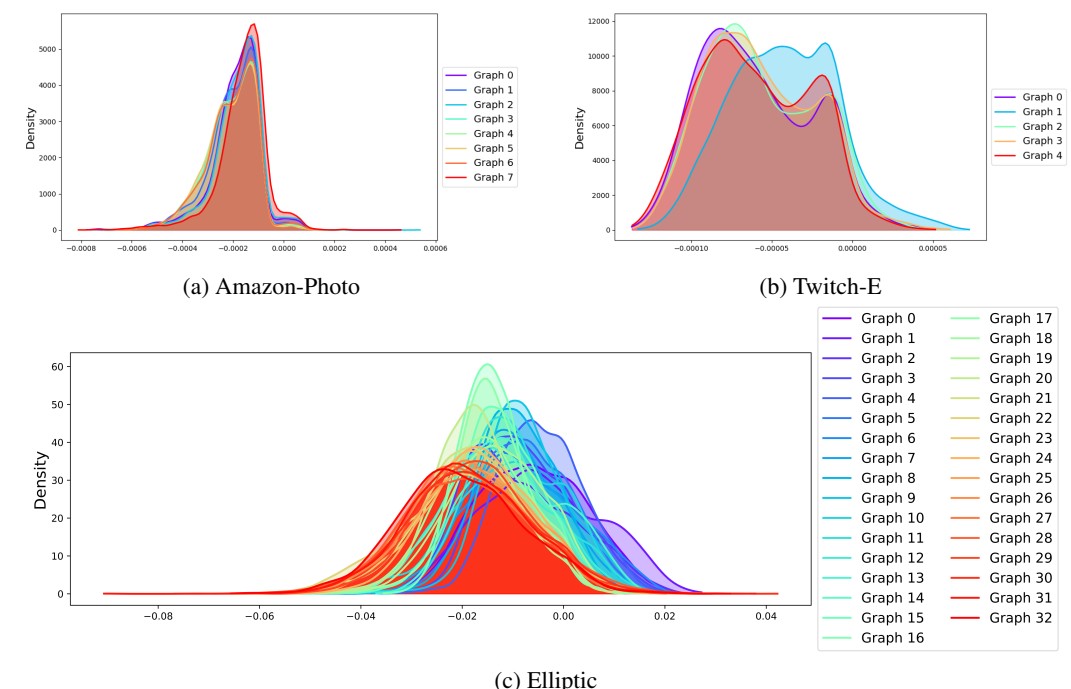

(a) Amazon-Photo

(b) Twitch-E

(c) Elliptic

Figure 9: OOD Distribution After Training on each dataset for OOD setting.

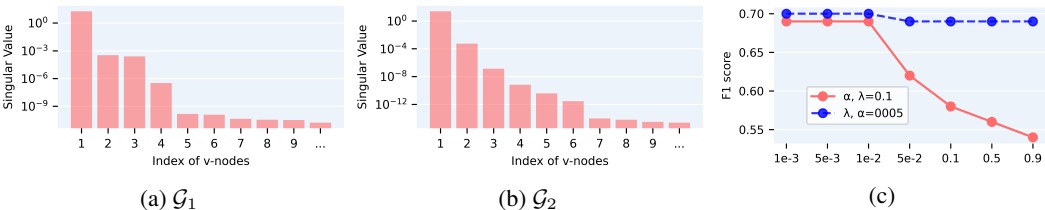

(a) $\mathcal{G}_1$

(b) $\mathcal{G}_2$

(c)

Figure 10: (a)(b) SVD of E in LROG After Training on graph $\mathcal{G}_1$, $\mathcal{G}_2$ in OGN-arXiv for OOD setting. (c) Parameter-performance curve of $\alpha/\lambda$ on Elliptic.

# F  ABLATION STUDY

## F.1  OPTIMIZATION OBJECT

In Figure 10, we show the parameter study of $\lambda$ and $\alpha$ in $\mathcal{L}_{A2A}$. Noted that there could be a different proportion of $\frac{\lambda}{(1-\alpha)}$, while it still should a rather value of $\alpha$ in that the constraint in Eq.(10)($\mathcal{L}_R$) should be satisfied first then the other objective could work.

## F.2  DIFFERENT AUGMENTATIONS METHODS USED IN OPTIMIZATION

In Target 12, we used DropEdge as the augmentation function $\mathcal{A}(\cdot)$ to obtain the augmented view. In practice, the choice of augmentation can be flexible and here we explore two other choices: node dropping and FlipEdge (You et al., 2020). Specifically, we adopt a ratio of 0.05 for node dropping, a ratio of 0.05 and 0.5 for FlipEdge, and ratios of 0.05 and 0.5 for DropEdge. We observe that (1) GOAT with any of the three augmentations can greatly improve the performance of GCN under distribution shift, and (2) different augmentations lead to slightly different performances on different datasets

### F.3 PARAMETERS IN LROG

After tuning over all datasets, the hyperparameter almost shows slightly different. Therefore, $d_{attn}$ is set to 8, $|n|$ is almost $10 \times C$ ($C$ is the class number of nodes in test graph). Furthermore, we explore that not alike the Transformer(Vaswani et al., 2017) in NLP, the multi-head attention and the residual connection cannot improve the performance in our LROG module, which indicates the graph structure data information learned with GNNs has different representation from those ones learned as in NLP as sequences.

## G MORE DISCUSSION

### G.1 LIMITATION

In this paper, we mainly focus on the OOD on the graph, while the idea in Eq.(12) could be expanded into more fields where neural networks learn a distribution. Secondly, due to computation limitations, we didn't conduct more experiments about the number of views in one epoch that could be shown in a figure. Finally, there could be a more relaxed optimization objective different from Eq.(Ap.9), we are willing to inspire more discussion and novel propositions.

### G.2 MORE EFFECT OF OOD ON GRAPH-STRUCTURE DATA

It is worth noting that the OOD problem in graph models can lead to significant risks and negative consequences in real-world applications. For instance, when GNNs are applied in financial risk control systems, distribution shifts in the input data may cause a large number of misjudgments, leading to severe economic losses or compliance issues. Similar risks exist in other high-stakes domains such as healthcare and criminal justice, where the reliability and robustness of graph-based decision-making systems under distributional changes are critical. Therefore, it is crucial to develop effective methods to detect and adapt to OOD scenarios in graph learning, and to carefully analyze and mitigate the potential negative societal impacts. Our work aims to contribute to this important research direction.

Another illustrative example of the potential negative impact of OOD issues in graph models is in the context of social network analysis for misinformation detection. GNNs have been widely adopted to identify fake news and rumors based on the propagation patterns and content features in social networks. However, the characteristics of misinformation can evolve rapidly over time, leading to distribution shifts between the training and test data. If the GNN-based misinformation detectors fail to adapt to such changes, they may miss emerging misinformation or cause false alarms, which can have severe societal consequences such as public panic, political manipulation, and erosion of trust in media. This urges the development of graph OOD detection and adaptation methods that can robustly handle the dynamic and adversarial nature of online misinformation. Our GOAT framework takes a step towards this goal by enabling test-time adaptation of GNNs to evolved data distributions.

