# OpenReview forum: "Map to Optimal: Adapting Graph Out-of-Distribution in Test Time"
_ICLR.cc/2025/Conference — Submitted to ICLR 2025_

### Official Review · Reviewer_32n6 · 2024-10-18

**Soundness:** 3
**Presentation:** 2
**Contribution:** 2
**Rating:** 5
**Confidence:** 4

**Summary:**

This paper introduces GOAT, a self-supervised test-time adaptation framework that improves the generalization of pre-trained GNNs on out-of-distribution (OOD) graph data. GOAT uses a graph augmentation strategy with a simple adapter to bridge the distribution gap between training and test data, enhancing performance on node classification tasks. It demonstrates superior results across various real-world OOD scenarios and benchmark datasets, while also being efficient in time and memory. The authors highlight GOAT's interpretability and its insights into handling distribution shifts in GNNs.

**Strengths:**

1. The paper emphasizes the interpretability of the OOD representations generated by GOAT, offering valuable insights into the model's adaptation process in out-of-distribution scenarios.
2. The paper presents a self-supervised strategy for adapting pre-trained GNNs to OOD scenarios without the need for labeled test data, which represents a significant practical contribution.
3. GOAT demonstrates superior computational and memory efficiency compared to several baseline methods, making it well-suited for large-scale graph datasets.

**Weaknesses:**

1. The method is primarily focused on node classification tasks. While it demonstrates promising results, its applicability to other graph-related tasks, such as graph classification or link prediction, has not been fully explored.
2. The paper does not provide a theoretical analysis to support the effectiveness of GOAT, relying instead on empirical evidence from experimental results.

**Questions:**

1. Is GOAT applicable to other graph tasks? If adjustments are required, are there any potential limitations?
2. Could a theoretical analysis or insights be provided to support the effectiveness of GOAT?

---

> ### Author Response · Authors · 2024-11-19
>
> Dear reviewer **32n6**,
>
> We appreciate your comments and your support for our work. We hope our response can address your concerns. Please find our detailed response below.
>
> **(Weakness 1 & Question 1)** Extended Tasks of our method
>
> We are happy to do more research in the future to improve our method so that it can be used in ***graph classification*** or ***edge prediction***.
> It should be noted that the test-time environment with OOD issue for graph classification may be different from that in node classification, because the distribution shift of graph classification may be more based on the graph level rather than the node level. Therefore, when generating augmented views, it may be necessary to consider whether to use a single test graph as an augmentation anchor or a batch of OOD graphs as the augmentation anchor. This will be a very interesting topic and future research direction.
>
> Moreover, our method can be extended to ***OOD detection***, as we showed in **Section 4.2**, after tuning the adaptor with our unsupervised loss $L_{\text{A2A}}$, the representation generated from our LROG module can be used as an indicator of the OOD degree of this test graph. After some special designs, it can also be used in ***knowledge graph*** ***completion*** to help entities have a better embedding, for unsupervised ***graph anomaly detection***, ***community detection***, etc.
>
> **(Weakness 2 & Question 2)** More theoretical analysis and insights
> As shown in **Fig 2**, a toy example, and **Appendix B**, our theoretical analysis demonstrates that adding input-level bias to graph node features can lead to better predictions within the decision boundary of pre-trained GNNs after feature aggregation and projection.
>
> The key insight of our approach lies in how we conceptualize and address the distribution shift problem in test-time graph adaptation. Our method can be viewed as adaptively adjusting the decision boundary of pre-trained GNNs according to test graph distributions. The design of our symmetric loss function with the L2 norm shares a profound connection with minimizing model variance in the test environment. However, as demonstrated in our ablation studies, optimizing this loss alone is insufficient for effective adaptation.
>
> The fundamental reason behind this observation is that the additional parameters introduced during training, which act as input-level bias, must maintain structural alignment with the GNN's feature extraction process. This motivation led to our design of the consistency loss, which is theoretically grounded in the mathematical concept of isomorphisms. As demonstrated in our two-view optimization formulation in **Appendix A**, while our relaxed optimization objective could potentially be replaced by alternative loss functions, two critical assumptions must hold:
>
> 1. The environment generating the graphs must be treated as a random variable
> 2. There exist one or multiple optimal graphs that enable the pre-trained GNN to achieve superior performance
>
> This theoretical framework provides a principled approach to adaptation while maintaining the structural information learned during pre-training. The interplay between symmetric and consistency losses ensures that the adaptation process respects both the global distribution alignment and local structural preservation.
>
> Thank you again for your comment. We hope our explanation can dispel your concerns. If you have any other questions or concerns, please feel free to let us know.

---

> > ### Comment · Reviewer_32n6 · 2024-12-03
> >
> > Thank you for the response. I have reviewed the comments and replies of the other reviewers. While the authors have addressed individual questions, I believe the paper requires further revision to provide a clearer and more cohesive explanation of their ideas. As the rebuttal phase is nearing its conclusion, I feel that the paper still needs additional refinement, and I cannot advocate for its acceptance in its current form.

---

### Official Review · Reviewer_bxGo · 2024-10-29

**Soundness:** 2
**Presentation:** 2
**Contribution:** 2
**Rating:** 5
**Confidence:** 4

**Summary:**

The paper proposes to improve the OOD performance of GNN tasks through a learnable adaptor. The adaptor is designed akin to the attention mechanism, and trained with a self-supervised loss that considers symmetric and consistent losses.

**Strengths:**

1. the method does not require the change of pre-trained weight.
2. the self-supervised loss considers the domain specific requirements regarding e.g., the symmetricity.
3. an explicit representation of the OOD as a matrix

**Weaknesses:**

1. all the ingredients of the method, including utilising attention mechanism for adaptation, the contrastive style supervised loss function, and using another branch (i.e., a learnable adaptor) for domain adaptation, are not new. The paper is a combination of several known techniques to work in a specific task.

2. I am not sure if this problem should be formulated as an OOD task, and it looks rather like a domain adaptation. The test time data is simply the data we may use in the new domain for adaptation. I assume for OOD tasks, we do not have test time data for learning the adaptor.

3. The performance against existing methods is not always better.

**Questions:**

Please see above the weaknesses.

---

> ### Author Response · Authors · 2024-11-19
>
> Dear reviewer **bxGo**,
>
> We appreciate your comments and your support for our work. We hope our response can address your concerns. Please find our detailed response below.
>
> **(Weakness 1)** Our contribution & More details of implementation.
>
> **Limitations of previous methods**
>
> - GTrans and the other methods with the direct modification on the graph’s edge require a hyperparameter to control the edge adjustment ratio, as considering all possible edge combinations would result in a combinatorial explosion due to the discrete search space over $N^{[0,1]}$ for each edge.
> - Previous test-time methods (GraphCTA, GraphTTA, and GTrans) rely on conventional contrastive losses in their self-supervised learning framework, which necessitates careful selection or design of positive and negative samples. For instance, GTrans specifically designs different data augmentation methods for node classification tasks to sample positive-negative pairs, where *positive samples* are generated through *DropEdge* while *negative samples* are obtained via *Node Shuffling* with specific augmentation parameters. Although this approach effectively captures discriminative features in the embeddings, it places excessive emphasis on embedding differences while *neglecting the crucial consistency (invariance) property* of the conditional distribution that underlies out-of-distribution shifts.
> - Although some train-time methods (EERM[1], MoleOOD[2]) utilize invariance learning by maximizing data variance in existing graph generation environments while minimizing their supervised losses, these methods require *supervised training* and consume *substantial computational resources.* These training-time approaches result in significant overhead in terms of both GPU memory consumption and training time.
>
> In contrast to these limitations, our work presents specific contributions that address each of these challenges one by one, as detailed in our key contributions:
>
> **Contribution**
>
> - **We achieve parameter efficiency through a low-rank adapter design.** Unlike methods that modify graph edges directly, our approach avoids combinatorial explosion by using a low-rank adapter structure. With our proposed adapter, we can efficiently compute global attention on large-scale graphs, enabling fast test-time tuning without the need for edge-ratio hyperparameters. The additional parameters learned by our adapter are directly applied as an input-level bias to the node features, offering an efficient mechanism for representation adaptation.
> - **We propose a novel consistency-aware framework** that goes beyond conventional contrastive learning. Our mathematical framework comprises Consistency Loss *$\mathcal{L}_{\text{con.}}$*, Regularization Loss *$\mathcal{L}_{\text{R}}$*, Symmetry Loss *$\mathcal{L}_{\text{symm.}}$*, and unified Augmentation-to-Augmentation Loss *$\mathcal{L}_{\text{A2A}}$*. This design explicitly addresses the consistency (invariance) property of conditional distributions in OOD settings, which was overlooked by previous test-time methods.
> - **We introduce GOAT**, an efficient test-time tuning paradigm that achieves consistent learning without the computational overhead of training-time methods. Our framework integrates a self-supervised loss mechanism with a low-rank adapter, enabling unlabeled test graphs to adapt to distribution shifts with minimal computational resources effectively. The method is theoretically grounded in a relaxed optimization objective, where learning across augmented views guides the additional parameters to optimize the fixed pre-trained parameters' performance on the shifted distribution.
>
> **(Weakness 2)** Problem Formulation & Why not Domain Adaptation.
>
> We want to clarify that while domain adaptation can be viewed as a specific case of OOD problems, our work addresses a broader scope. In OOD scenarios, we may deal with data from the same domain, such as cases in *OGB-ArXiv and Cora,* but with different training and testing distributions, which is precisely our case.
>
> We aim to present a unified framework for handling various distribution shifts in test-time graph adaptation. In our setting, each test graph is processed individually at test time using a pre-trained GNN model, where the test graphs are generated from different environments, leading to OOD challenges.
>
> Importantly, we only have access to the graph data without corresponding labels during the adaptation process. Our experimental validation encompasses three real-world scenarios: *artificial transformations, domain adaptation, and temporal evolution*. Additionally, we provide comprehensive comparisons between our method and existing domain adaptation approaches in **Table 7** & **Table 8**.
>
> *(Please check the following comment for more responses)*

---

> > ### Author Response · Authors · 2024-11-19
> >
> > **(Weakness 3)** Significance of our experiment.
> >
> > Our experimental results in **Table 2.** demonstrate strong statistical significance and substantial performance improvements. Specifically, **out of 24** experimental settings, our method achieves **statistically significant** improvements (validated by t-tests) **in 19 cases**, showing overwhelming advantages over the ERM baseline. Moreover, when compared with GTrans, the current state-of-the-art approach, our method achieves comparable or superior performance in 14 different settings. The magnitude of improvement is particularly noteworthy - our method outperforms GTrans by a considerable margin, achieving scores of **67.92 vs. 63.04**(*Elliptic*) and **54.20 vs. 51.27**(*FB100*) in the most significant cases.
> >
> > Thank you again for your comment. We hope our explanation can dispel your concerns. If you have any other questions or concerns, please feel free to let us know.

---

### Official Review · Reviewer_2CzR · 2024-10-31

**Soundness:** 2
**Presentation:** 1
**Contribution:** 3
**Rating:** 3
**Confidence:** 3

**Summary:**

When using GNNs, it may be that graphs used at test time (or under deployment) are different in systemic and meaningful ways from the graphs the GNN was trained on due to changes in the data-generating environment, meaning test-time data may be too far out-of-distribution to be effectively classified by the GNN.  This paper introduces GOAT, an approach to modifying OOD test-time graphs so they can be accurately classified by a pre-trained GNN.

GOAT transforms the graph taking into account three things.  First, the transformed graph and original graph (? - see Questions) must perform well under some self-supervised task. Second, this is regularized by encouraging the transformed graph and original graphs to have similar GNN outputs ("Symmetry").  Third, the approach is "consistent," where the transformation of the GNN output on the original graph is encouraged to be similar to the GNN output of the transformation of the graph.

Performance on test-time graph classification is compared against other approaches, and GOAT is shown to outperform or compete closely with the other techniques.  An ablation study is done to show that each of the three portions are necessary.

**Strengths:**

This is an appropriately important question, and state-of-the-art performance on these tasks is important.  The paper shows that the approach works, and runs quickly and with little memory usage.  This is the foundation of an appropriately impactful paper.

**Weaknesses:**

- Overall, things are poorly explained, and I do not understand the approach well.  A few specific examples of these points of confusion follow here and in the Questions portion, but throughout, the storytelling needs to be clearer.  In particular, the intuitive explanations of Eqs 2, 8, and 10 are insufficient for the main contributions of the paper.
- Quite a bit of important notation is insufficiently explained (see questions).
- Wording in Assumption 1 is poor grammar and confusing. "Environment is the condition that generates graph".  "Environment" seems to be a vague idea of "things are changing, so the graphs are changing," but the environments are used quite mathematically.  For your approach, what makes an environment suddenly become the next environment in the sequence?
- The experimental results do show improvement, but not overwhelmingly so.  I don't feel these results earn the benefit of the doubt on confusing (to me) explanations.

Overall, I definitely do not understand key portions of this paper.  It is always possible this is my fault, but I believe in this case, it is due to poor and uncareful explanations.

**Minor thing**
- Repeated misspelling of "Augmentation to augmentation" with "augmentaion"

**Questions:**

- In Equation 2, what is $ \hat G_{te} $ ?  Without an explanation of $\hat G_{te}$ and a differentiation from $G_{te}$ in line 161, the equation 2 makes little sense, making a poor foundation for the rest of the loss functions.
- What is $\textbf{H}$ on lines 211 and 240?  What is a "layer" in this context on line 240?
- How are $\textbf{K'}$ and $\textbf{V'}$ on line 243 learned?  Are they just linear projections?  The use of ``learned" suggests to me something more complex is happening, but it's never explained?
- Where do $W_Q, W_K, and W_V$ come from in the section containing equation 6?
- What is $L_R$ in Eqn 12?  Is that Eqn 2?
- On line 277, what makes $G_v$ augmented?  As written, aren't those just being drawn from the probability distribution natively?  Is some notation missing?  What are $p$ and $q$ on line 278?

---

> ### Author Response · Authors · 2024-11-19
>
> Dear reviewer **2CzR**,
>
> We appreciate your comments and your support for our work. In the following response, we have carefully addressed each point raised and hope to resolve any uncertainties.
>
> **(Weakness 1)** Our Contribution & Contrast to Previous Methods.
>
> **Limitations of previous methods**
>
> - GTrans and the other methods with the direct modification on the graph’s edge require a hyperparameter to control the edge adjustment ratio, as considering all possible edge combinations would result in a combinatorial explosion due to the discrete search space over [0,1] for each edge.
>
> - Previous test-time methods (GraphCTA, GraphTTA, and GTrans) rely on conventional contrastive losses in their self-supervised learning framework, which necessitates careful selection or design of positive and negative samples. For instance, GTrans specifically designs different data augmentation methods for node classification tasks to sample positive-negative pairs, where positive samples are generated through DropEdge while negative samples are obtained via Node Shuffling with specific augmentation parameters. Although this approach effectively captures discriminative features in the embeddings, it places excessive emphasis on embedding differences while neglecting the crucial consistency (invariance) property of the conditional distribution that underlies out-of-distribution shifts.
>
> - Although some train-time methods (EERM[1], MoleOOD[2]) utilize invariance learning by maximizing data variance in existing graph generation environments while minimizing their supervised losses, these methods require supervised training and consume substantial computational resources. These training-time approaches result in significant overhead in terms of both GPU memory consumption and training time.
>
> In contrast to these limitations, our work presents specific contributions that address each of these challenges one by one, as detailed in our key contributions:
>
> **Contribution**
>
> - **We achieve parameter efficiency through a low-rank adapter design.** Unlike methods that modify graph edges directly, our approach avoids combinatorial explosion by using a low-rank adapter structure. With our proposed adapter, we can efficiently compute global attention on large-scale graphs, enabling fast test-time tuning without the need for edge-ratio hyperparameters. The additional parameters learned by our adapter are directly applied as an input-level bias to the node features, offering an efficient mechanism for representation adaptation.
> - **We propose a novel consistency-aware framework** that goes beyond conventional contrastive learning. Our mathematical framework comprises Consistency Loss *$\mathcal{L}_{\text{con.}}$*, Regularization Loss *$\mathcal{L}_{\text{R}}$*, Symmetry Loss *$\mathcal{L}_{\text{symm.}}$*, and unified Augmentation-to-Augmentation Loss *$\mathcal{L}_{\text{A2A}}$*. This design explicitly addresses the consistency (invariance) property of conditional distributions in OOD settings, which was overlooked by previous test-time methods.
> - **We introduce GOAT**, an efficient test-time tuning paradigm that achieves consistent learning without the computational overhead of training-time methods. Our framework integrates a self-supervised loss mechanism with a low-rank adapter, enabling unlabeled test graphs to effectively adapt to distribution shifts with minimal computational resources. The method is theoretically grounded in a relaxed optimization objective, where learning across augmented views guides the additional parameters to optimize the fixed pre-trained parameters' performance on the shifted distribution.
>
> **(Weakness 3)** More explanation of the environment in Assumption 1
>
> In our assumption, the “Environment” $\mathrm{e}$ (as formally defined by Wu et al. [1] and Yang et al.[2] ) is treated as a random variable, which means it can be learned through back-propagation. The changes in the environment cause OOD phenomena in test graphs. This is why we aim to represent the 'Environment' to address graph OOD issues at test time. The visualization of our learned environment can be seen in **Fig 5(c)** and **Fig 9**.
>
> **(Weakness 4)** Significance of our experiment.
>
> Our experimental results in **Table 2.** demonstrate strong statistical significance and substantial performance improvements. Specifically, **out of 24** experimental settings, our method achieves **statistically significant** improvements (validated by t-tests) **in 19 cases**, showing overwhelming advantages over the ERM baseline. Moreover, when compared with GTrans, the current state-of-the-art approach, our method achieves comparable or superior performance in 14 different settings. The magnitude of improvement is particularly noteworthy - our method outperforms GTrans by a considerable margin, achieving scores of **67.92 vs. 63.04**(*Elliptic*) and **54.20 vs. 51.27**(FB100) in the most significant cases.
>
> (Please check the following comment for more responses)

---

> > ### Author Response · Authors · 2024-11-19
> >
> > **(Questions 1 & 6)** $\hat{\mathcal{G}}_{te}$ in Eq.2 and how $G_v$  on line 227 ‘augmented’
> >
> > As we showed **on line 161**, $\hat{\mathcal{G}}_{te}$ , theoretically, is the graph sampled from the distribution of the test-time environment generates the test graph, therefore the environment can be varied.
> >
> > In practice, according to **Assumption 2**, in test time, the test graph is a single graph. $\mathcal{G}_v \sim p(\mathrm{G} | \mathrm{e} = e_i)$ can be sampled by DropEdge, Flipedge, Subgraph sampling, or other augmentation methods.
> >
> > **(Question 2)** $\mathrm{H}$ on lines 211 and 240
> >
> > $H^{k} \in \mathbb{R}^{N \times d_k}$ , on line 240, is the node representations embedded by the pre-trained k-layer GNN. To elaborate, one layer includes aggregation and the linear projection with a non-linear activation function.
> >
> > **(Questions 3 & 4)**  $\mathrm{K’}$  and $\mathrm{V’}$ on line 243, $W_Q$, $W_K$, and $W_V$
> >
> >  $\mathrm{K’}$  and $\mathrm{V’}$ $\in \mathbb{R}^{|n| \times N}$ are two learnable matrices that are initialized to be full rank along the |n| dimension. Changing the term “learned” to “learnable” in the original text would indeed help avoid this confusion. We appreciate you bringing this to our attention.
> >
> >  $W_K$, $W_V$ $\in \mathbb{R}^{d_k \times d_{\text{attn}}}$ , and $W_Q \in \mathbb{R}^{d_0 \times d_{\text{attn}}}$ are also random initialized learnable weight matrices.
> >
> > **(Question 5)** What is $\mathcal{L}_\text{R}$
> >
> > As shown on **lines 312** and **313**, the left term of **Eq.11** is $\mathcal{L}_\text{R}$
> >
> > **(Question 6)** What are p and q on line 278
> >
> > Both p and q are integers ranging from 1 to $|v|$ (the number of sampled $G_v$), where we use different variables p and q to explicitly indicate that they represent two different sampled graphs, rather than the same graph as in **Eq 10.** Our notation aims to emphasize that these are drawn from separate sampling processes.
> >
> > **(Minor thing)** We appreciate you pointing out the spelling errors. We will fix the original manuscript and will conduct thorough proofreading during the revision process to ensure the highest quality of presentation.
> >
> > Thank you for pointing out these issues. We will review all symbols and further define them in the final version. We hope our explanation can dispel your concerns. If you have any other questions or concerns, please feel free to let us know.
> >
> > Ref:
> >
> > [1] Wu, Qitian, et al. "Handling Distribution Shifts on Graphs: An Invariance Perspective." *International Conference on Learning Representations*.
> >
> > [2] Yang, Nianzu, et al. "Learning substructure invariance for out-of-distribution molecular representations." *Advances in Neural Information Processing Systems* 35 (2022)

---

> > > ### Comment · Reviewer_2CzR · 2024-11-26
> > >
> > > Thank you for the response.  I found the writing confusing enough that I would need a new draft to raise my score, rather than just individual answers to questions.  I hope to see an improved paper in a later venue, but I cannot advocate its acceptance in its current form.

---

### Official Review · Reviewer_nBtR · 2024-11-03

**Soundness:** 2
**Presentation:** 2
**Contribution:** 2
**Rating:** 3
**Confidence:** 4

**Summary:**

This paper addresses the challenge of out-of-distribution (OOD) graph data at test time and proposes a test-time adaptation method, called GOAT, for graph neural networks. The key idea is to capture the condition/environment that generates graphs. A low-rank adapter generates representations by which the test graph’s node features are modified to align with the training graph environment. The adapter is optimized using a self-supervised loss function that enforces symmetry and consistency between the different augmented views of the test graph. Empirical results on six benchmark datasets show the effectiveness of GOAT in handling various OOD scenarios.

**Strengths:**

1. Introducing a low-rank adapter is a computationally efficient solution for large graphs.

2. The authors offer a continuous formulation of the graph test-time adaption problem, as well as the optimization process.

3. The idea of enforcing symmetry and consistency between the different augmented views of the test graph is new.

**Weaknesses:**

1. The proposed method shares many similarities with GTrans. Although this paper continuously formulates the problem, we can see from the discrete version in the Appendix that these two methods both consider modifying the node attribute by a representation that could catch the out-of-distribution drift. Both employ dropEdge as a sampling method for contrastive learning.  In fact, GTrans modifies both the node attributes and graph structures. The main differences lie in the low-rank adapter and the loss function.

2. The technical novelty is somewhat limited. The low-rank adapter is a straightforward application of existing techniques. The self-supervised loss function is a common solution for test time adaption for graphs.

3. In the problem formulation, the authors did not explain why the optimal parameter that minimizes the expected supervised loss could be obtained from optimizing the point estimation problem. This is a key problem since Y_te is not available at test time.

**Questions:**

1. In Eq. (2), what $\hat{\cal G}_{te}$  means?

2. In Proposition 1, the statement that "modifies the graph structure within the learned parameter space of a pre-trained GNN model" is not clear. Could you explain how the graph structure is modified and how it relates to the learned parameter space?

3. The instance on Page 5 of what LROG learns is not a substantial example to illustrate what LROG can learn. How LROG could capture the environment change? As well as Fig. 3, which is illustrative but does not provide the idea of why LROG could catch both the change in graph structure and feature distribution shift.

4. In Section 3.3, the expression $g_\psi(f(\cal G)) \approx f(g_\psi(\cal G))$  is inaccurate because the input of $g_\psi$ is a graph, not the GNN's output.

5. In Table 2, the average rank may not be a fair comparison as different datasets have varying performance gaps, especially there are 94.35 vs 94.32, 94.79 vs 94.76, and 55.83 vs 55.82

---

> ### Author Response · Authors · 2024-11-19
>
> Dear reviewer **nBtR**,
>
> We appreciate your comments and your support for our work. We hope our response can address your concerns. Please find our detailed response below.
>
> **Weakness：**
>
> **(Weakness 1 & 2)** Main Contribution & Differences with GTrans
>
> **Limitations of previous methods**
>
> - GTrans and the other methods with the direct modification on the graph’s edge require a hyperparameter to control the edge adjustment ratio, as considering all possible edge combinations would result in a combinatorial explosion due to the discrete search space over $N^{[0,1]}$ for each edge.
> - Previous test-time methods (GraphCTA, GraphTTA, and GTrans) rely on conventional contrastive losses in their self-supervised learning framework, which necessitates careful selection or design of positive and negative samples. For instance, GTrans specifically designs different data augmentation methods for node classification tasks to sample positive-negative pairs, where *positive samples* are generated through *DropEdge* while *negative samples* are obtained via N*ode Shuffling* with specific augmentation parameters. Although this approach effectively captures discriminative features in the embeddings, it places excessive emphasis on embedding differences while *neglecting the crucial consistency (invariance) property* of the conditional distribution that underlies out-of-distribution shifts.
> - Although some train-time methods (EERM[1], MoleOOD[2]) utilize invariance learning by maximizing data variance in existing graph generation environments while minimizing their supervised losses, these methods require *supervised training* and *consume substantial computational resources.* These training-time approaches result in significant overhead in terms of both GPU memory consumption and training time.
>
> In contrast to these limitations, our work presents specific contributions that address each of these challenges one by one, as detailed in our key contributions:
>
> **Contribution**
>
> - **We achieve parameter efficiency through a low-rank adapter design.** Unlike methods that modify graph edges directly, our approach avoids combinatorial explosion by using a low-rank adapter structure. With our proposed adapter, we can efficiently compute global attention on large-scale graphs, enabling fast test-time tuning without the need for edge-ratio hyperparameters. The additional parameters learned by our adapter are directly applied as an input-level bias to the node features, offering an efficient mechanism for representation adaptation.
> - **We propose a novel consistency-aware framework** that goes beyond conventional contrastive learning. Our mathematical framework comprises Consistency Loss *$\mathcal{L}_{\text{con.}}$*, Regularization Loss *$\mathcal{L}_{\text{R}}$*, Symmetry Loss *$\mathcal{L}_{\text{symm.}}$*, and unified Augmentation-to-Augmentation Loss *$\mathcal{L}_{\text{A2A}}$*. This design explicitly addresses the consistency (invariance) property of conditional distributions in OOD settings, which was overlooked by previous test-time methods.
> - **We introduce GOAT**, an efficient test-time tuning paradigm that achieves consistent learning without the computational overhead of training-time methods. Our framework integrates a self-supervised loss mechanism with a low-rank adapter, enabling unlabeled test graphs to effectively adapt to distribution shifts with minimal computational resources. The method is theoretically grounded in a relaxed optimization objective, where learning across augmented views guides the additional parameters to optimize the fixed pre-trained parameters' performance on the shifted distribution.
>
> **(Weakness 3)**  Why optimizing the point estimation problem can minimize the expected supervised loss?
>
> The connection between point estimation and supervised loss minimization can be explained through our theoretical framework:
>
> 1. According to **Proposition b** in **Appendix A**, for any test graph $\mathcal{G} \sim p(\mathrm{G}|\mathrm{e} = e_i)$, there exists an optimal OOD representation $E^*$ that maps the test graph to the distribution where the pre-trained GNN performs optimally, thus $G^*$.
> 2. Our formulation of point estimation aims to find this optimal mapping through the augmentation-to-augmentation strategy. Specifically:
>     - The symmetric loss ensures the consistency between different views of the same test graph
>     - The consistency loss maintains the structural alignment with GNN's feature extraction
>     - The regularization term prevents degenerate solutions
>
> As proved in **Appendix A**, under constraint *$E[f(A'_2, X'_2 + E^{\*}_2) - f(A'_2, X'_2)] = 0$*, the unsupervised objective ($P_{A2A}$) becomes equivalent to the supervised objective ($P_{A2S})$.
>
> Therefore, by optimizing our point estimation objective, we are effectively minimizing an upper bound of the expected supervised loss without requiring access to labels."
>
> (Please check the following comment for more responses)

---

> ### Author Response · Authors · 2024-11-19
>
> **(Question 1)** $\hat{\mathcal{G}}_{te}$ in Eq.2
>
> As we show on **line 161**, $\hat{\mathcal{G}}_{te}$ is the graph sampled from the test time environment with *DropEdge*, *FlipEdge*, *subgraph sampling*, or other augmentation methods.
>
> **(Question 2)** “modifies the graph structure within the learned parameter space of a pre-trained GNN model  $f_{\theta^*}$” in Proposition 1
>
> In **Proposition 1**, $g_\psi$ is designed to operate beyond individual graph instances - it works with samples observed and collected from the environment generating the test graph. When we refer to “modifying the graph structure within the learned parameter space of $f_{\theta^*}$”, we are specifically highlighting how this function influences the GNN's learning process: it enables more accurate aggregation paths and information flow during node feature embedded by  $f_{\theta^*}$. Notably, in our approach, rather than directly modifying the graph structure, we add learned bias (generated by our low-rank adapter) to node features. Through the GNN's message-passing mechanism, these biased node features effectively guide the structural information flow toward more accurate directions, achieving implicit graph structure modification.
>
> **(Question 3)** Why does the module LROG catch both the change in graph structure and feature distribution shift?
>
> It is because the input of the LORG module maintains information on the graph edge changes as the embedding of the node feature is actually aggregated by the pre-trained GNN network.
>
> Our LROG module implements a cross-attention mechanism where:
>
> 1. Query (Q) is derived from raw input node features
> 2. Key (K) and Value (V) are obtained from node embeddings that have undergone GNN's aggregation and non-linear transformation, thus inherently encoding edge information through message passing
>
> This design, coupled with our optimization objectives, allows us to simultaneously monitor and align both node and edge-level characteristics between the test graph and the pre-trained GNN's encoded knowledge, capturing both their discrepancies and consistencies.
>
> **(Question 4)** $g_\psi(f\mathcal(G))\approx f(g_\psi\mathcal(G))$
>
> We respectfully highlight that we defined the combination of the $g$   and $f$ in **footnote 4** **line 322**:
>
> **(Question 5)** Fair Comparison
>
> Thank you for your careful observation of the ranking comparison. We acknowledge that the absolute performance differences in some cases are small. However, we would like to clarify that:
>
> 1. The average ranking is computed by first ranking methods within each dataset independently, then averaging these ranks across datasets. This approach helps normalize the varying scales of performance across different datasets.
> 2. More importantly, our method shows consistent improvements across diverse scenarios - from artificial transformations to temporal evolution, and across different backbones. The consistency of improvement, rather than just the magnitude, demonstrates the robustness of our approach.
> 3. Furthermore, on challenging datasets like Elliptic and OGB-ArXiv where distribution shifts are more severe, our method shows more substantial improvements (e.g., **67.92 vs 63.04** on *Elliptic with SAGE backbone*, **54.20 vs. 51.27** on *FB100 with GAT backbone*). It can be said that our method achieves comparable or superior performance to GTrans.
>
> We hope our explanation can dispel your concerns. If you have any other questions or concerns, please feel free to let us know.

---

### Meta-Review · Area_Chair_wRah · 2024-12-23

**Metareview:**

This paper introduces GOAT, a self-supervised test-time adaptation framework for graph neural networks (GNNs) in out-of-distribution (OOD) settings. GOAT leverages a low-rank adapter and a consistency-aware loss to align test graphs with the training distribution. The reviewers acknowledged the importance of addressing OOD challenges and the computational efficiency of the method but raised concerns about limited novelty, inadequate theoretical grounding, and unclear presentation. While the authors provided detailed responses, additional experiments, and revised explanations, these efforts were insufficient to fully address the reviewers' major concerns.

**Additional Comments On Reviewer Discussion:**

The discussion phase primarily revolved around the method's novelty, theoretical contributions, and clarity. The reviewers questioned the distinction between GOAT and prior methods, such as GTrans, and noted the lack of a strong theoretical basis to support the proposed framework. The authors responded by emphasizing the efficiency of their low-rank adapter, conducting additional experiments to validate the design choices, and revising the manuscript to improve clarity. However, reviewers remained unconvinced about the method’s incremental contribution and its broader applicability to tasks beyond node classification. While the rebuttal addressed some questions, the paper was not deemed ready for acceptance due to unresolved issues in presentation and impact.

---

### Decision · Program_Chairs · 2025-01-22

Reject